# Finding Regions of Heterogeneity in Decision-Making via Expected Conditional Covariance

**Justin Lim**[*]
MIT CSAIL and IMES
Cambridge, MA
justinl@mit.edu

**Christina X Ji**[*]
MIT CSAIL and IMES
Cambridge, MA
cji@mit.edu

**Michael Oberst**[*]
MIT CSAIL and IMES
Cambridge, MA
moberst@mit.edu

**Saul Blecker**
NYU Langone
New York, NY
saul.blecker@nyulangone.org

**Leora Horwitz**
NYU Langone
New York, NY
leora.horwitz@nyulangone.org

**David Sontag**
MIT CSAIL and IMES
Cambridge, MA
dsontag@csail.mit.edu

## Abstract

Individuals often make different decisions when faced with the same context, due to personal preferences and background. For instance, judges may vary in their leniency towards certain drug-related offenses, and doctors may vary in their preference for how to start treatment for certain types of patients. With these examples in mind, we present an algorithm for identifying types of contexts (e.g., types of cases or patients) with high inter-decision-maker disagreement. We formalize this as a causal inference problem, seeking a region where the assignment of decision-maker has a large causal effect on the decision. Our algorithm finds such a region by maximizing an empirical objective, and we give a generalization bound for its performance. In a semi-synthetic experiment, we show that our algorithm recovers the correct region of heterogeneity accurately compared to baselines. Finally, we apply our algorithm to real-world healthcare datasets, recovering variation that aligns with existing clinical knowledge.

## 1 Introduction

Understanding heterogeneity in decision-making is an established problem in medicine (Birkmeyer et al., 2013; Corallo et al., 2014; De Jong et al., 2006), consumer choice (Ortega et al., 2011; Scarpa et al., 2005), and law (Kang et al., 2012; Kleinberg et al., 2018; Arnold et al., 2018). In the context of medicine, this is referred to as the study of practice variation (Atsma et al., 2020; Cabana et al., 1999), where it is often observed that doctors, facing the same clinical context, will make different decisions. Likewise, in a legal context, judges often differ in their leniency in their decisions regarding bail (Kleinberg et al., 2018), juvenile incarceration (Aizer and Doyle Jr, 2013), the use of alternatives to incarceration (Di Tella and Schargrodsky, 2013), and incarceration length (Kling, 2006). In some scenarios this variation may be justified: The best medical treatment may not be obvious. In others, it may be grossly unfair, as in the case of racial bias in bail decisions (Arnold et al., 2018).

---

[*]Equal contribution

35th Conference on Neural Information Processing Systems (NeurIPS 2021).

In this work, we tackle the question of how to find and characterize this variation in the first place. In particular, we present a learning algorithm for identifying a "region of heterogeneity", defined as a subset of all contexts (e.g., patients, cases) for which the identity of the decision-maker substantially affects the decision. In medicine, a better understanding of treatment variation can inform the development and dissemination of clinical guidelines. In the legal domain, characterizing the cases where judges vary most in their leniency may help with investigating potential issues of fairness.

We formalize characterizing the region of heterogeneity as a causal inference problem: We want to characterize examples where *changing the decision-maker* would have resulted in a different decision. The challenge is two-fold: First, we only observe a single decision-maker per example, so we cannot directly observe how (for instance) multiple judges would have decided the same case. Second, our data on individual decision-makers is often scarce. For instance, in Section 5, we consider a medical dataset with more than 400 doctors, each of whom has fewer than 9 samples on average.

We will refer to decision-makers as "agents", and our contributions are as follows: In Section 2, we propose an objective defined in terms of counterfactual decisions across different agents, and show that this objective can be identified from observational data. Moreover, this objective does not require the use of agent-specific statistical models, making it amenable to our sparse setting. In Section 3, we give an iterative algorithm to identify regions of disagreement by maximizing this objective and provide intuition (in the form of a generalization bound) for the factors that drive its performance. In Section 4, we use a semi-synthetic dataset, derived from crowd-sourced recidivism predictions, to demonstrate that our algorithm recovers the correct region of heterogeneity accurately, even when there are many agents. Finally, in Section 5, we apply our algorithm to a real-world healthcare dataset and confirm that it recovers intuitive regions of variation in first-line diabetes treatment. We conclude with a discussion of related work and implications. Our code is available at https://github.com/clinicalml/finding-decision-heterogeneity-regions.

Our algorithm does not determine whether variation is inherently good or bad or how it should be addressed. Rather, more careful study with domain experts would be required to determine if variation can (or should be) reduced and how. In addition, false discovery of variation is possible and could have a negative impact. We expect that validation on independent datasets would be required in real-world applications, using the regions identified by our method as plausible hypotheses to test.

## 2 Characterizing Heterogeneity from a Causal Perspective

### 2.1 Notation

Let the data be drawn from a distribution $\mathbb{P}(X, A, Y)$, where $X$ is a random variable representing context (or features), $A$ is a discrete agent, and $Y \in \{0, 1\}$ is the binary decision. The spaces of all $X$ and $A$ are denoted as $\mathcal{X}$ and $\mathcal{A}$ with realized values as lower case $x$ and $a$, respectively. Indicator variables $\mathbf{1}[\cdot]$ are one if the statement inside the brackets is true and zero otherwise. For a subset $S \subseteq \mathcal{X}$, $\mathbf{1}[x \in S]$ is sometimes written as a function $S(x)$, where $S : \mathcal{X} \to \{0, 1\}$. A subset $S$ may have several disjoint regions. $\mathbb{E}[Y|X \in S]$ denotes the average $Y$ across samples in $S$. For instance, if $S = \{X : X_0 < 10\}$, then $\mathbb{E}[Y|X_0 < 10]$ is a scalar average of $Y$ among samples with $X_0 < 10$.

### 2.2 Heterogeneity as a Causal Contrast

Our conceptual goal is to identify a region $S \subseteq \mathcal{X}$ where different agents tend to make different decisions even when faced with the same context. We can formalize this in the language of potential outcomes from the causal inference literature (Pearl, 2009; Hernán and Robins, 2020), which for clarity we will refer to as *potential decisions*: In particular, we denote $Y(a)$ to be the potential decision of agent $a$. The fundamental challenge of causal inference is that we do not observe all potential decisions $\{Y(a) : a \in \mathcal{A}\}$ for each sample, but only a single decision $Y$. With this in mind, we will make the following assumptions, standard in the literature on causal effect estimation.

**Assumption 1** (Causal Identification Assumptions). *(i) Consistency: $Y = y, A = a \implies Y(a) = y$, and (ii) No Unmeasured Confounding (NUC): For all $a \in \mathcal{A}$, $Y(a) \perp\!\!\!\perp A \mid X$.*

Consistency links the potential $Y(a)$ to the observed $Y$, and NUC says that there are no unobserved factors that influence both the assignment of agents and the decision itself. For instance, the quasi-random assignment of cases to judges conditioned on features $X$ satisfies NUC (Kleinberg et al., 2018).

NUC may be violated if key aspects of the case (e.g., misdemeanor vs. felony) are omitted as features. For instance, misdemeanor and felony cases may be seen by different judges and have different decision processes, but this variation is not due to agent preferences. Given these assumptions, we propose a causal measure of agent-specific bias, defined as a contrast between potential decisions.

**Definition 1** (Conditional Relative Agent Bias). For an agent $a \in \mathcal{A}$ and a subset $S \subseteq \mathcal{X}$, the conditional relative agent bias is defined as

$$\mathbb{E}[Y(a) - Y(\pi(x)) \mid A = a, X \in S] \tag{1}$$

where $Y(a)$ is the potential decision of agent $a$, and $Y(\pi(x)) := \sum_{a'} \mathbb{E}[Y(a') \mid x]\pi(a' \mid x)$ denotes the expected potential decision under the agent assignment distribution $\pi(a' \mid x) := \mathbb{P}(A = a' \mid X = x)$.

Note that under Assumption 1, $Y(\pi(x)) = \mathbb{E}[Y \mid x]$,[2] but here we emphasize its causal interpretation as the expected decision of a random agent. Equation (1) represents the relative difference between the decision of an agent (on their particular distribution of cases in the region) and the potential decision of a random agent. In particular, Equation (1) can be written as follows under Assumption 1

$$\mathbb{E}[Y(a) - Y(\pi) \mid A = a, X \in S] = \int_{x \in S} \mathbb{E}[Y(a) - Y(\pi) \mid X = x] p(x \mid A = a, X \in S) dx, \tag{2}$$

where we shorten $Y(\pi(x))$ to $Y(\pi)$. This is the average difference (over $p(x \mid a)$, restricted to those $x$ in the set $S$) of the conditional expected difference between $Y(a)$ and $Y(\pi)$. For example, suppose that the agent $a$ is a judge who is particularly lenient on bail decisions for felony arrests (the region $S$), and $Y = 1$ denotes granting bail. Then imagine the following counterfactual: Take the felony cases that are assigned to this judge and reassign each individual case, described by $x$, to a random judge $a'$, proportionally to $p(a' \mid x)$. We may then observe, on average, that the bail rate would decrease, because most judges are less lenient than judge $a$, corresponding to a positive value of Equation (1).

Equation (1) has the additional advantage of being easy to estimate: Under Assumption 1, it can be rewritten[3] as $\mathbb{E}[Y - \mathbb{E}[Y \mid X] \mid A = a, X \in S]$, the expected residual in predicting (using the conditional expectation $\mathbb{E}[Y \mid X]$) the decision of an agent $a$ across the context $x$ typically seen by that agent.

## 2.3 Formalizing a Causal Objective

Our primary goal is to discover a region $S$ where substantial heterogeneity exists across agents. To do so, we define an aggregate objective across a group $G$ of agents, where $G(a) \in \{0, 1\}$ is an indicator function for membership.

$$Q(S, G) := \sum_{a:G(a)=1} \mathbb{P}(A = a \mid X \in S)\mathbb{E}[Y(a) - Y(\pi) \mid A = a, X \in S], \tag{3}$$

We now show that this quantity can be identified and estimated from observational data without requiring agent-specific statistical models, before discussing the interpretation of this objective.

**Theorem 1** (Causal Identification). *Under Assumption 1, $Q(S, G)$ can be identified as*

$$Q(S, G) = \mathbb{E}_S[Cov(Y, G \mid X)] = \mathbb{E}_S[(Y - \mathbb{E}[Y \mid X])G], \tag{4}$$

*where $\mathbb{E}_S[\cdot] := \mathbb{E}[\cdot \mid X \in S]$ and $Cov(Y, G \mid X)$ is the conditional covariance.*

Theorem 1 and other theoretical results are proven in Appendix A. The result follows from proving that the agent-specific bias (Definition 1) is identifiable using the expected conditional covariance between $Y$ and the binary indicator $\mathbf{1}[A = a]$. With this in mind, we optimize the following objective, where the set $S$ is constrained to be at least a certain size $\beta$ and $\mathcal{S}$ is a hypothesis class of functions $S$.

$$\max_{S \in \mathcal{S}, G} Q(S, G) \text{ s.t., } \mathbb{P}(S) \geq \beta, \tag{5}$$

**Interpretation**: Intuitively, this objective measures the disagreement between the agents in the group $G(a) = 1$ and the overall average $\mathbb{E}[Y \mid X]$ on the region $S$. Hence, the choice of group is important for interpreting the objective: If $G(a) = 1$ for all agents, the objective will be zero for any set $S$, as can be seen from Equation (4), applying the definition of the conditional expectation.

---

[2]See Proposition A1 for a short proof, and Proposition A2 for the derivation of Equation (2).
[3]See Proposition A3 in Appendix A.1.

Accordingly, we seek a region $S$ for which the partially maximized objective $L(S) := \max_G Q(S, G)$ is large: This partial maximization is obtained by taking $G(a) = 1$ whenever the conditional relative agent bias of agent $a$ (on the set $S$) is non-negative. Thus, Equation (5) can be re-written as

$$\max_G Q(S, G) = \sum_{a \in \mathcal{A}} \mathbb{P}(A = a \mid X \in S) \, |\mathbb{E}[Y(a) - Y(\pi) \mid A = a, X \in S]|_+ \,, \tag{6}$$

where $|x|_+ := \max(x, 0)$, and this objective becomes an average over agents who have a positive bias. This population objective is also equivalent (up to a constant factor) to the (weighted) sum of the magnitude of each agent's conditional relative agent bias. See Proposition A4 in Appendix A.1.

**Lack of Overlap**: We have *not* made the overlap or positivity assumption that $\mathbb{P}(A = a \mid x) > 0$ for all $x, a$. While this assumption is required to identify conditional causal effects $\mathbb{E}[Y(a) - Y(a') \mid X]$ (Nie and Wager, 2017; Wager and Athey, 2018; Shalit et al., 2017), it is not required for identifying our causal contrast. Our problem only requires each context has a positive probability of being seen by more than one decision maker. For instance, suppose that $S$ contains both misdemeanors and felonies and there are four judges $a_0, \ldots, a_3$. If judges $a_0$ and $a_1$ make bail decisions exclusively for felonies while judges $a_2$ and $a_3$ make bail decisions exclusively for misdemeanors, our measure captures disagreement between $a_0$ and $a_1$ and between $a_2$ and $a_3$ even though comparisons between $a_0$ and $a_2$ or other pairs are impossible to make. Thus, we have chosen to compare $Y(a)$ to the decisions of *viable alternative agents*, weighted by their probability $p(a' \mid x)$ of being selected for such a case.

## 3 Identifying Regions with Heterogeneity

In Section 3.1, we introduce an iterative optimization algorithm for a finite sample version of Objective (5) that alternatingly optimizes $S$ and $G$. In Sections 3.2 and 3.3, we discuss practical heuristics for choosing the region size parameter $\beta$ on training data and validating if the resulting region generalizes to held-out data. Finally, we build intuition for the factors that influence performance of this algorithm via a generalization bound in Section 3.4 under simplifying assumptions.

### 3.1 Iterative Optimization Algorithm

We let $\hat{Q}(S, G)$ be the empirical analog of $Q(S, G)$ (Equation 4), which we can write as follows,

$$\hat{Q}(S, G) := \frac{1}{\sum_{a,j} \mathbf{1}\left[x_{aj} \in S\right]} \sum_{a,j} (y_{aj} - f(x_{aj})) \cdot G(a) \cdot \mathbf{1}\left[x_{aj} \in S\right]. \tag{7}$$

where $f(x)$ denotes a model of the conditional expectation $f(x) \approx \mathbb{E}[Y \mid X = x]$. For simplicity of notation, we assume that there are $R$ samples (indexed by $j$) for each of a finite set of $N$ agents (indexed by $a$), giving $N \cdot R$ samples in total.

Our algorithm (Algorithm 1) takes as input the data $\{(x_{aj}, y_{aj})\}$ and a minimum region size $\beta$, and outputs a model $h(x)$ and a threshold value $b$ that describe a region of heterogeneity $S = \{x \in \mathcal{X}; h(x) \geq b\}$. Starting with $S = \mathcal{X}$ (the entire space), the algorithm identifies the grouping that maximizes $\hat{Q}(S, G)$, then uses that grouping to identify the region maximizing the same quantity, repeating this process until convergence. The algorithm uses a classifier $f(x)$ to estimate $\mathbb{E}[Y \mid X = x]$ and a regression model $h(x)$ to estimate the conditional covariance of the decision $Y$ and the grouping $G$ at $X = x$. Note that we can use any supervised learning algorithms for $f$ and $h$, allowing us to learn interpretable regions as part of the algorithm if $h(x)$ is interpretable (e.g., decision trees). If sufficient data is available, samples can be split into three parts for estimating $f(x)$ in line 2, computing $G(a)$ in lines 5-8, and training $h(x)$ and estimating the $(1 - \beta)$-th quantile in line 10. We do not perform this sample splitting because our sample sizes are already small. Under-fitting $f(x)$ by further restricting the sample size could lead to false discovery if $f(x)$ does not capture the variation explained by $X$.

**Optimizing over $G$ given $S$.** Given a region $S$, our first result identifies the grouping $G : \mathcal{A} \to \{0, 1\}$ that maximizes $\hat{Q}(S, G)$ and shows that it can be expressed in terms of $\hat{Q}(S, \mathbf{1}[A = a])$.

**Proposition 1.** *Given $S \subseteq \mathcal{X}$, $\hat{Q}(S, G)$ is maximized over the space of functions $G : \mathcal{A} \to \{0, 1\}$ at $G_S$, where $G_S(a) = \mathbf{1}\left[\hat{Q}(S, \mathbf{1}[A = a]) \geq 0\right]$.*

Intuitively, this proposition states that to maximize the empirical expected conditional covariance of the decision and grouping on a region, we must group agents by whether their residuals $y_{aj} - f(x_{aj})$ are (on average) positive or negative on $S$.

---

**Algorithm 1** Identifying regions with variation

---

1: **Input:** Data $\{\{x_{aj}, y_{aj}\}_{j=1}^{R}\}_{a=1}^{N}$, minimum region size $\beta$.
2: Fit a model $f(x)$ to $\mathbb{E}(Y \mid X = x)$.
3: Initialize $S = \mathcal{X}$.
4: **repeat**
5:     **for** $a = 1, \ldots, N$ **do**
6:         Compute $\hat{Q}(S, \mathbf{1}[A = a])$, where $\hat{Q}(S, \mathbf{1}[A = a]) \coloneqq \frac{1}{\sum_j \mathbf{1}[x_{aj} \in S]} \sum_j (y_{aj} - f(x_{aj})) \mathbf{1}[x_{aj} \in S]$,
7:         Set $G(a) = 1$ if $\hat{Q}(S, \mathbf{1}[A = a]) \geq 0$ and 0 otherwise.
8:     **end for**
9:     Compute $b_{aj} = (y_{aj} - f(x_{aj}))G(a)$, $a = 1, \ldots, N$, $j = 1, \ldots, R$.
10:     Fit a model $h(x)$ to predict $b_{aj}$ from $x_{aj}$, and let $b$ be the $(1 - \beta)$-th quantile of $h(x_{aj})$.
11:     $S' \leftarrow S$.
12:     $S \leftarrow \{x_{aj}; h(x_{aj}) \geq b\}$.
13: **until** $S = S'$ or iteration limit reached.
14: **Output:** Model $h$ and threshold $b$, defining a region $S = \{x \in \mathcal{X}; h(x) \geq b\}$.

---

**Optimizing over $S$ given $G$.** To optimize $\hat{Q}(S, G)$ for a fixed grouping $G$ over the hypothesis class $\mathcal{S}$, we train a model $h(x)$ to predict $(y_{aj} - f(x_{aj}))G(a)$ given $x_{aj}$, where $h \in \mathcal{H}$. Using $h$ as an estimate in Eq. 7, we find a set $S$ to maximize the quantity $\frac{1}{\sum_x \mathbf{1}[x \in S]} \sum_{x \in S} h(x)$, the empirical conditional expectation of $h(x)$ over $S$. This quantity is maximized (subject to our $\beta$ constraint) by including the largest $\beta$-fraction of the $h(x_{aj})$ in $S$. Hence, we pick $b$ as the $(1 - \beta)$-th quantile of $h(x_{aj})$ and choose our region as $\hat{S}_G = \{x \in \mathcal{X}; h(x) \geq b\}$.

## 3.2 Tuning the Region Size Parameter

For real datasets, we need to choose $\beta$ without knowledge of the true value. Given that our objective can be calculated on held-out data using the functions $S, G$, a seemingly obvious approach would be to compute $Q(S, G)$ on a validation set and select the $\beta$ that leads to the highest $Q(S, G)$. However, for a fixed data distribution, smaller values of $\beta$ tend to produce higher values of $Q(S, G)$, and there is a trade-off between finding a smaller region of higher variation and a larger region that may include areas of lower (but still meaningful) variation. This motivated our original constraint $\mathbb{P}(S) \geq \beta$.

To select $\beta$, we propose a heuristic inspired by permutation-based hypothesis testing (Wasserman, 2004). We compare the training objective to a reference distribution of values (for the same $\beta$) that might be seen *even if all agents followed the same policy*. For each candidate $\beta$, we (i) run our algorithm and compute the objective on training data $q_{\text{obs}} \coloneqq \hat{Q}(\hat{S}, \hat{G})$. (ii) For $T$ iterations, we randomly shuffle the agents and re-run the algorithm to get a new objective value. This gives us a distribution $\hat{\mathbb{P}}_{\text{null}}$ over $Q(S, G)$ from a distribution where $\mathbb{P}(X, Y)$ and $\mathbb{P}(A)$ are unchanged but $Y, X \perp\!\!\!\perp A$. (iii) Finally, we compute a p-value $p_\beta = \hat{\mathbb{P}}_{\text{null}}(Q > q_{\text{obs}})$ and choose the $\beta$ with the smallest p-value. In Section 4.2, we find that this heuristic empirically recovers the true $\beta$ value in semi-synthetic settings.

## 3.3 Validation of the Region

We may wish to validate the learned region $\hat{S}$ independently of the grouping $\hat{G}$. In particular, finding $\hat{G}$ is not our main goal, and we observe in our semi-synthetic experiments that our algorithm can find the true region $S$ even when the grouping $\hat{G}$ is fairly poor (due to few samples per agent), as shown in Appendix B.2. We can optimize over $G$ in $Q(S, G)$ to obtain an objective that depends only on $S$ and can be used to compare regions. By Proposition 1, we obtain an empirical analog of Equation (6) as

$$\hat{L}(S) \coloneqq \max_G \hat{Q}(S, G) = \frac{1}{\sum_{a,j} \mathbf{1}[x_{aj} \in S]} \sum_a \left| \sum_j (y_{aj} - f(x_{aj})) \mathbf{1}[x_{aj} \in S] \right|_+, \tag{8}$$

where $|x|_+$ is equal to the positive part $[x]_+ = \max(x, 0)$ as before. We then use this objective $\hat{L}(S)$ to answer the following question: Does our chosen region $\hat{S}$ yield a significantly higher objective value on test data than a randomly selected region of the same size? An example of this analysis is given in Table 1 for the real-data experiment in Section 5.

## 3.4 Generalization Error

We give a generalization bound for Algorithm 1 to build intuition for the factors that influence performance. To derive this bound, we consider a simplified setting, where there exists a set $S', G'$ such that the following set of assumptions hold.

**Assumption 2** (Group-based variation). *For all $x \in S'$, $\mathbb{E}[Y \mid X = x, A = a] = \mathbb{E}[Y \mid X = x, G'(a)]$ and for all $x \notin S'$, $\mathbb{E}[Y \mid X = x, A = a] = \mathbb{E}[Y \mid X = x]$*

**Assumption 3** (Non-zero relative biases). *There exists a constant $\alpha > 0$ such that for all $x \in S'$,*

$$\mathbb{E}[Y \mid X = x, G'(A) = 1] - \mathbb{E}[Y \mid X = x] > \alpha, \quad and \quad \mathbb{E}[Y \mid X = x, G'(A) = 0] - \mathbb{E}[Y \mid X = x] < -\alpha,$$

**Assumption 4** (All agents see samples in $S'$). *There exists a constant $\omega > 0$, such that for every $a \in \mathcal{A}$, $\mathbb{P}(X \in S' \mid A = a) > \omega \mathbb{P}(X \in S')$.*

Note that under these assumptions, $S', G'$ maximize the objective $Q(S, G)$ (see Appendix A.4), so we will refer to them as $S^*, G^*$ for the remainder of this section. Assumption 2 says that there are two groups of agents, who follow two distinct decision policies within a region $S^*$ but follow an identical decision policy outside of $S^*$. Assumption 3 says that one group has a positive bias across all of $S^*$, relative to the average over both groups, and the other group has a negative bias. To simplify the analysis, we also make Assumption 4 that every agent has some non-zero chance of observing some contexts $X$ in the region, but note that we do not require that $p(x \mid a) > 0$ for all $x \in S'$. Under these assumptions, we demonstrate that the first iteration of Algorithm 1 will find, with high probability, a region $\hat{S}$ whose value $Q(\hat{S}, G^*)$ (for the same grouping $G^*$ defined above) is close to that of the optimal $S^*$. Note that we do not claim that the iterative algorithm finds the globally optimal solution. For simplicity, we assume that $f(x)$ perfectly recovers $\mathbb{E}[Y \mid X]$. This can be relaxed at the cost of additional terms in the bound that go to zero as the overall sample size increases. Under Assumptions 2, 3, and 4, assume that $\mathbb{P}(S^*) = \beta$. For the informal version presented here, we assume that $\mathbb{P}(S^*) = \mathbb{P}(\hat{S}) = \beta$, where $\hat{S}$ is returned by our algorithm, and that exactly a $\beta$-fraction of our samples fall into $S^*$ and $\hat{S}$ (in the Appendix, we give a version without these simplifications).

**Theorem 2.** *Under the assumptions above, if $S^* \in \mathcal{S}$ and $R > \frac{2 \ln 2}{\alpha^2 \beta^2 \omega^2}$, the first iteration of Algorithm 1 returns $\hat{S}$ such that, with probability at least $1 - \delta$, $Q(S^*, G^*) - Q(\hat{S}, G^*) \leq \epsilon$, where*

$$\epsilon = \sqrt{\frac{2 \ln(3/\delta)}{\beta N \cdot R}} + \frac{2}{\beta} \left( \eta + \sqrt{\frac{3\eta(1 - \eta)}{\delta \cdot N}} \right) + \frac{1}{\beta} \left( 2\mathcal{R}(\mathcal{S}, N \cdot R) + 4\sqrt{\frac{2 \ln(12/\delta)}{N \cdot R}} \right),$$

*where $\mathcal{R}(\mathcal{S}, N \cdot R)$ is the Rademacher complexity of $\mathcal{S}$, and $\eta := \exp\left(-R\alpha^2\beta^2\omega^2/2\right)$.*

The term $\eta$ plays an important role: It bounds the expected misclassification error $\mathbb{P}(\hat{G}(a) \neq G^*(a))$. For sufficiently large $R$, we have that $\eta < 1/2$ with high probability, i.e., we have a better-than-random chance of identifying the group for an individual agent. Moreover, $\eta$ decreases as we increase the number of samples $R$ for each agent, the separation $\alpha$ between the two groups on $S^*$, the region size $\beta$, and the constant $\omega$. For sufficiently small $\eta$, our algorithm discovers a region whose value (in terms of $Q(S, G^*)$) is close to that of the optimal region. The generalization bound improves as the number of agents $N$ increases, the number of samples $R$ for each agent increases, or the complexity of the hypothesis class decreases. The latter is measured here by the Rademacher complexity $\mathcal{R}(\mathcal{S}, N \cdot R)$ of our hypothesis class $\mathcal{S}$, which can be bounded by standard arguments. In conclusion, under some additional assumptions, Algorithm 1 identifies an approximately optimal solution with high probability after one iteration. We show via semi-synthetic experiments in Appendix B.3 that convergence is generally fast in practice.

## 4 Semi-Synthetic Experiment: Recidivism Prediction

For conceptual motivation in the introduction, we discussed the legal system: As a potential application of our method, one could determine types of cases for which the idiosyncratic preferences of judges have a significant impact on their decisions. Lacking data on judge decisions with sufficient context, we turn to the more controlled setting of human predictions of recidivism.

**Dataset:** We use publicly available data from Lin et al. (2020),[4] who ask participants on Amazon's Mechanical Turk platform to make recidivism predictions based on information present in the

---

[4]Available at `https://github.com/stanford-policylab/recidivism-predictions`

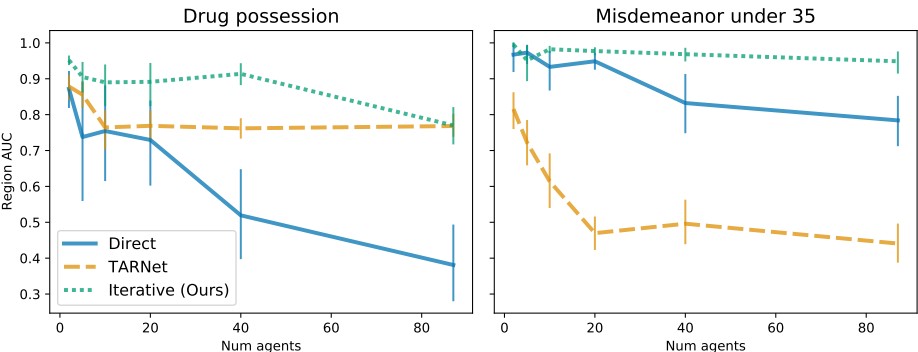

Figure 1: Comparison of our method and best baselines at identifying region of heterogeneity, as measured by the held-out test AUC for classifying samples into the true region of heterogeneity. Total number of samples is fixed. Baselines are described in Section 4.1. Uncertainty bands give 95% intervals for the mean derived via bootstrapping over 10 random seeds using seaborn (Waskom, 2021). Left: Region is modelled using a ridge regression in the drug possession semi-synthetic set-up. Right: Region is modelled using a random forest for the misdemeanor under age 35 set-up.

"Correctional Offender Management Profiling for Alternative Sanctions" (COMPAS) dataset for Broward County, FL (Dressel and Farid, 2018). Participants (or "agents") are shown 5 risk factors: age, gender, number of prior convictions, number of juvenile felony charges, and number of juvenile misdemeanor charges. The charge in question is also given, as well as whether the charge is a misdemeanor or felony. The dataset contains 4550 cases evaluated by 87 participants.

**Semi-Synthetic Policy Generation:** To benchmark our method, we generate semi-synthetic data where we have access to a "ground truth" region of heterogeneity. We retain the features presented to the original participants and construct two policies, which we refer to as the "base" and "alternative" policies: For the base policy, we learn a logistic regression model on the binary decisions across the whole dataset. For the alternative policy, an extra positive term is added to the logistic regression for samples within the region. We construct two scenarios with different regions of variation: (1) all drug possession charges, and (2) all misdemeanor charges where the individual is 35 years old or younger. These make up 22% and 21% of the data, respectively. Then, we generate synthetic agents (randomly assigned to cases) and assign half of the agents to the base policy and half to the alternative. Synthetic decisions are then sampled from the logistic regressions. For each scenario, the two groups of agents follow the same stochastic policy outside of the region, and one group systematically prefers $Y = 1$ within the region. More details can be found in Appendix B.1.

### 4.1 Performance versus Baselines

**Baselines:** We compare how well our approach identifies the true region of heterogeneity with several baselines. To our knowledge, the problem of finding regions of heterogeneity with a large number of agents has not been studied before. Many causal inference methods for treatment effect estimation are designed for a single, binary treatment. However, naively estimating the treatment effect between each pair of providers would scale $O\left(|\mathcal{A}|^2\right)$. Therefore, we develop new baselines. Some (including the causal forest and U-learner adaptations described in Appendix B.2) are based on causal inference methods augmented to identify a region of heterogeneity and grouping of agents where possible.

*Direct models:* This baseline measures how much adding the agent to the feature set improves prediction of decisions. We fit logistic regressions with and without the agent feature to estimate $\mathbb{E}[Y \mid A, X]$ and $\mathbb{E}[Y \mid X]$. For each sample $(x, y, a)$, we compute $|y - \mathbb{E}[Y \mid X = x]| - |y - \mathbb{E}[Y \mid A = a, X = x]|$ to quantify how much the model with agents outperforms the model without agents. Then, we fit a "region model" to predict this quantity from $X$. This region model is either a ridge regression, decision tree, or random forest model. Finally, we compute the top $\beta$ quantile of predictions from the region model in the training and validation sets and use this cut-off to select points in the test set.

*TARNet:* A treatment-agnostic representation network (Shalit et al., 2017) models the outcomes of all treatments for each sample by learning a shared representation of the sample features and then

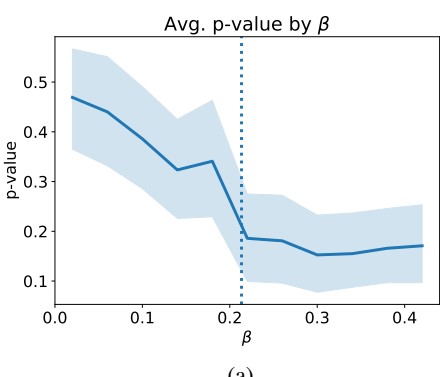
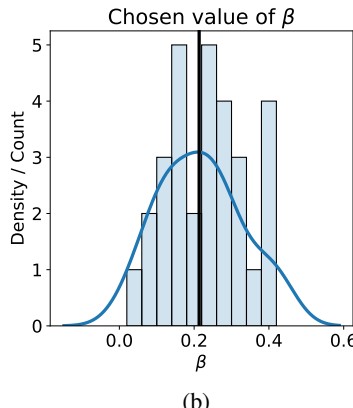

Figure 2: Results of tuning $\beta$ over 30 semi-synthetic datasets. (a) Average p-value for each candidate $\beta$, with 95% uncertainty intervals for the mean generated by bootstrapping. The dotted vertical line represents the true value of $\beta$. (b) Distribution of $\beta$ with the smallest p-value over the datasets.

having a separate prediction head for each treatment. We implement the shared representation model using a 2-layer neural network with ReLU and dropout. Each prediction head is a linear layer with a sigmoid function. TARNet predicts $\mathbb{E}[Y \mid X, A]$ for every agent for each sample. $\mathrm{Var}_A[\mathbb{E}[Y \mid X, A]]$ measures the variation across all agents if they had seen context $X$ and is analogous to our objective without grouping. We predict this quantity with the region models as in the direct model baseline.

**Results:** We evaluate how well the algorithms identify the samples within the region of heterogeneity when we vary the number of agents among 2, 5, 10, 20, 40, and 87, where 87 is the number of agents in the original dataset. Figure 1 shows the best overall region models for each set-up, with the other models deferred to Appendix B.2. The metric in Figure 1 is the region AUC, defined as how well the model classifies whether samples belong in the region of heterogeneity when compared to the true region. Algorithm 1 consistently performs well for both semi-synthetic set-ups, especially when the number of agents increases to a realistic level (and the number of samples per agent decreases). The direct baseline deteriorates very rapidly as the number of agents increases in the drug possession set-up, while the TARNet baseline deteriorates rapidly in the misdemeanor under age 35 set-up. Refer to Appendix B.2 for additional baseline details, region models, and evaluation metrics. We also show that our method is robust in a set-up with more than 2 agent groups in Appendix B.4.

### 4.2 Tuning the Region Size

We validate the proposed approach of tuning $\beta$ (discussed in Section 3.2), by applying the methodology to our semi-synthetic setting here. We sample 30 semi-synthetic datasets and consider candidate values of $\beta$ in $[0.02, 0.42]$ in increments of 0.04. For each proposed value of $\beta$ we use $T = 40$ random permutations of the agents. Figure 2 presents results for the misdemeanor under age 35 set-up. As the candidate value of $\beta$ increases (up to the true value of $\beta$), the p-value decreases, and the distribution of selected $\beta$ values are centered on the true value of $\beta$.

## 5 Real-data Experiment: First-Line Diabetes Treatment

We apply our algorithm on a real-world dataset consisting of first-line (initial) treatment for type 2 diabetes and examine how the treatment variation we discover aligns with clinical knowledge. We present an additional real-world experiment (on Parkinson's disease) in Appendix D.

**Data and Setup**: We use an observational, de-identified, dataset provided by a large health insurer in the United States, spanning from 2010 to 2020. The task is to classify first-line treatment decisions between metformin ($Y = 0$)–the typical recommendation from the American Diabetes Association– and other common first-line treatments such as sitagliptin, glipizide, glimepiride, or glyburide ($Y = 1$) (American Diabetes Association, 2010; Hripcsak et al., 2016). As relevant clinical features, we include the patients' most recent eGFR (mL/min/1.73m2) and creatinine (mg/dL) measurements,

Table 1: Objective values $L(S)$ for the learned region on the training and test datasets, along with the distribution of values for randomly generated regions $S_{rand}$ given as mean (standard deviation).

| Metric | Subset | Value |
|---|---|---|
| $L(\hat{S})$ | Train | 0.1029 |
| $L(\hat{S})$ | Test | 0.0924 |
| $L(S_{rand})$ | Test | 0.0507 (0.0073) |

incidence of heart failure, and treatment date. Because treatment date does not define a type of patient, we omit it from the region model. However, including it in the outcome model is essential because of increasing use of metformin over time. The agent indicator $A$ is the group practice of the doctor responsible for the patient's treatment. 3,980 patients and 447 group practices are included in our cohort. After requiring at least 4 patients per agent, 3,576 patients and 176 group practices are included. This filter ensures each group practice has at least 1 sample in the training and validation sets and at least 2 samples in the test set. In this experiment, we choose $\beta = 0.25$ as input to our algorithm. See Appendix C for additional cohort definition and set-up details.

**Interpretation of Results**: To interpret the region, we use decision trees as our region model $h(x)$. The tree is visualized in Appendix C. The decision tree identifies the region of heterogeneity as the union of (i) eGFR below 71.5 and (ii) eGFR above 98.5 and creatinine above 0.815. These regions align with clinical intuition. In the first region, low eGFR values indicate impaired renal function (Group et al., 2009), which is a contraindication for metformin since it is traditionally associated with increased risk of lactic acidosis (Tahrani et al., 2007). Still, treatment decisions can vary here because guidelines for managing patients with eGFR below 45 are lacking (Group et al., 2015). Note that this region provides an example of how our algorithm works when overlap is not satisfied. Although 33 of 176 group practices do not see patients with these features, we can still conclude that this is a region of heterogeneity among the 143 agents with cases. In the second region, there are no obvious contraindications for metformin. Thus, understanding why some doctors on average only prescribe metformin 78% of the time to patients in this region may help us identify whether this is a region in which we can standardize practice.

**Assessing Significance**: In Table 1, we perform a sanity check, assessing whether our algorithm identifies a region $S$ whose variation in held-out data is higher than that of a randomly selected region, using the partially optimized objective $L(S) = \max_G Q(S, G)$ laid out in Section 3.3 to compare regions directly. Table 1 shows that $L(\hat{S})$ is similar on the training and test data. Furthermore, we compare $L(\hat{S})$ on the test set to the distribution of $L(S_{rand})$, where $S_{rand}$ are random subsets of the test data of the same size as $\hat{S}$. We compute the latter distribution using 100 random subsets and observe that the test value of $L(\hat{S})$ is more than two standard deviations above the mean of the latter. This gives us confidence that the discovered region $S$ generalizes to a region of heterogeneity beyond the training set. We direct the reader to Appendix C for additional analyses, such as evaluating the stability of the region over multiple folds of splitting the data.

# 6 Related Work

Beyond previously mentioned connections to causal effect estimation, we highlight a few areas of research that share similar goals to our own.

**Agent-specific Models of Decisions**: Prior works have modeled agent-specific decision-making processes by estimating a separate model for each agent. Abaluck et al. (2016) model heterogeneity in physician tendency to run diagnostic tests. Chan Jr et al. (2019) estimate radiologist skill based on diagnosis and miss rates. In our setting, unlike in diagnosis, there is no "correct" decision that can be incorporated into the model. Ribers and Ullrich (2020) assume there is provider-specific noise in determining patient type, which affects the pay-off functions for deciding whether to prescribe antibiotics. When only a few decisions are observed per agent, these agent-specific models cannot be estimated reliably. Currie and MacLeod (2017) also incorporate physician beliefs and skill into a pay-off function. They estimate an aggregate logistic choice model (for C-sections) across all physicians and then learn how individual physicians deviate from this model. They do not learn the regions where this deviation occurs, as they focus on how heterogeneity is associated with

downstream outcomes. Norris (2020) looks for disagreement between agents but relies on some cases being seen by multiple agents. We assume each case is seen by only one decision-maker.

**Conditional Independence Testing**: While our objective maximizes a causal notion of dependence, one could instead ask if $Y$ is conditionally independent of $A$ given $X$. Many metrics exist for testing conditional independence, such as the Hilbert-Schmidt independence criterion (HSIC) (Fukumizu et al., 2007; Zhang et al., 2012), conditional mutual information (Runge, 2018), conditional correlation (Ramsey, 2014), and expected conditional covariance (Shah and Peters, 2020). We give the last a causal interpretation under some assumptions and seek a region that maximizes it, in lieu of testing.

**Hierarchical / Mixture Models**: Bayesian methods are often used to estimate group-level effects, such as a per-physician effect on patient outcomes (Tuerk et al., 2008), where group identifiers are included as a categorical feature in a multi-level generalized linear model (Gelman and Hill, 2006). Alternatively, one could assume a conditional mixture model (Bishop, 2006), where agents belong to latent clusters that each have their own policy. However, both of these methods require parametric assumptions on the distribution of $\mathbb{P}[Y \mid x, a]$, and even so, the optimal mixture model is not necessarily identifiable when both the clusters and policies are unknown (Dasgupta and Schulman, 2007). By contrast, our method does not require making particular parametric assumptions about $\mathbb{E}[Y \mid x, a]$ and seeks to learn the region of heterogeneity directly.

**Feature Evaluation**: Checking for heterogeneity can also be framed as feature evaluation, where we would like to evaluate whether adding the agent identifier will increase predictive power. Feature evaluation methods typically maximize dependence between selected features and labels, utilizing measures similar to those in conditional independence testing, such as the HSIC (Song et al., 2007). Other methods use the correlation of the new feature with the loss gradient of the current predictor as a measure of utility (Koepke and Bilenko, 2012). In contrast, we focus not on checking marginally for the predictive power of agent identifiers, but rather identifying a region.

**Crowdsourcing**: Our work is conceptually related to identifying which samples are difficult to label via crowdsourcing annotations (Karger et al., 2014; Whitehill et al., 2009). Most crowdsourcing models are generative with latent variables for the correct sample labels, sample difficulty, agent expertise, and agent bias. They then optimize for the likelihood of the observed labels. The set of difficult samples is analogous to our region of heterogeneity. The main difference with our problem is that we do not require any notion of the "correct" label.

## 7  Discussion

In this work, we take a causal perspective on finding regions where agents (i.e., decision-makers) have heterogeneous preferences, formalizing this heterogeneity in terms of counterfactual (or "potential") decisions across agents. We propose a causal measure of agent-specific biases and give an objective that aggregates this bias over agents. This objective can be identified from observational data and written in terms of an expected conditional covariance. Importantly, for our applications, this does not require building agent-specific models or assuming overlap across all agents.

We give an iterative algorithm to find a region that maximizes this objective. Then, we demonstrate in semi-synthetic experiments that our algorithm accurately recovers the region of heterogeneity and scales well with the number of agents. In contrast, performance of baslines deteriorates when the number of agents increases. Although our experiments have low-dimensional spaces, we hypothesize our algorithm would scale well to high-dimensional feature spaces since the average policy and region models can handle high-dimensional input spaces. Finally, on a real-world medical dataset, we show that our algorithm can yield insights that align with clinical knowledge.

Our work is motivated by understanding variation in human decision-making. In the judicial domain, our method may help unearth types of cases where decisions are highly dependent on the judge assigned to the case. In the medical domain, our approach may identify types of patients where new guidelines may be required to help doctors make decisions more consistent with standard of care. Domain expertise is required to determine the implications of the regions discovered by our method. Beyond these applications, we see our approach as a useful data science tool for understanding heterogeneity in decisions that appears to be driven by individual-level preferences.

**Acknowledgements**: We would like to thank Aaron Smith-McLallen, James Denyer, Luogang Wei, Johnathon (Kyle) Armstrong, Neil Dixit, Aditya Sai, and the rest of the data science group at Independence Blue Cross, whose expertise, data, and support enabled the diabetes experiment. We would also like to thank Rebecca Boiarsky for converting the diabetes data to the OMOP common data model format, Monica Agrawal for her helpful comments, and other members of the lab for insightful discussions. We are also grateful to Charles Venuto and Monica Javidnia for their advice on Parkinson's. This work was supported in part by Independence Blue Cross, Office of Naval Research Award No. N00014-17-1-2791, an Abdul Latif Jameel fellowship, and a NSF CAREER award.

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
