# Appendix

This appendix contains the following sections:

- **Proofs**: Section A contains proofs of theoretical results presented in the main paper.
- **Semi-Synthetic Experiment: Additional Details**: Section B contains additional details and experimental results for the semi-synthetic experiment presented in Section 4.
- **Diabetes Experiment: Additional Details**: Section C contains additional details and experimental results for the real-data diabetes experiment presented in Section 5.
- **Additional Real-Data Experiment: Parkinson's**: Section D contains an additional real-data experiment, on a medical dataset of patients with Parkinson's disease.

Most of our experiments were run on CPUs, with only the TARNet baseline run on a GEForce GTX GPU. We estimate the compute time to be on the order of 100 hours.

# A   Proofs of Theoretical Results

## A.1   Proof of Minor Claims in Section 2

**Proposition A1.** *Under the assumptions of consistency and no-unmeasured-confounding (NUC), the following equivalence holds*

$$Y(\pi(x)) = \mathbb{E}[Y \mid x],$$

*where $Y(\pi(x))$ is defined as $\sum_{a'} \mathbb{E}[Y(a') \mid X = x]\pi(a' \mid x)$, where $\pi(a' \mid x) \coloneqq \mathbb{P}(A = a' \mid X = x)$.*

*Proof.* Based on the definition of $Y(\pi(x))$, we can write it as follows, using the fact that $Y$ is binary.

$$
\begin{aligned}
Y(\pi(x)) &\coloneqq \sum_a \mathbb{P}(Y(a') = 1 \mid X = x)\pi(a' \mid x) \\
&= \sum_{a'} \mathbb{P}(Y(a') = 1 \mid A = a', X = x)\mathbb{P}(A = a' \mid X = x) && \text{(NUC)} \\
&= \sum_{a'} \mathbb{P}(Y = 1 \mid A = a', X = x)\mathbb{P}(A = a' \mid X = x) && \text{(Consistency)} \\
&= \sum_{a'} \mathbb{P}(Y = 1, A = a' \mid X = x) \\
&= \mathbb{E}[Y \mid X = x]
\end{aligned}
$$

$\square$

**Proposition A2.** *Under the assumptions of consistency and no-unmeasured-confounding (NUC), the following equivalence holds*

$$\mathbb{E}[Y(a) - Y(\pi) \mid A = a, X \in S] = \int_{x \in S} \mathbb{E}[Y(a) - Y(\pi) \mid x]p(x \mid A = a, X \in S)dx,$$

*Proof.* This can be seen as follows, noting that by our definition of $Y(\pi(x))$ as a function of $x$, $\mathbb{E}[Y(\pi(x)) \mid X = x] = Y(\pi(x))$

$$
\begin{aligned}
\mathbb{E}[Y(a) - Y(\pi) \mid A = a, X \in S] &= \int_{x \in S} \mathbb{E}[(Y(a) - \mathbb{E}[Y \mid x]) \mid A = a, X = x]p(x \mid a, X \in S)dx \\
&= \int_{x \in S} (\mathbb{E}[Y(a) \mid X = x] - \mathbb{E}[Y \mid x])\, p(x \mid a, X \in S)dx && \text{(NUC)} \\
&= \int_{x \in S} \mathbb{E}[Y(a) - Y(\pi) \mid x]p(x \mid A = a, X \in S)dx
\end{aligned}
$$

$\square$

**Proposition A3.** *Under the assumptions of consistency and no-unmeasured-confounding (NUC), the conditional relative agent bias can be written as*

$$\mathbb{E}[Y(a) - Y(\pi) \mid A = a, X \in S] = \mathbb{E}[Y - \mathbb{E}[Y \mid X] \mid A = a, X \in S]$$

*Proof.* By Propositions A1 and A2, we can re-write the conditional relative agent bias as

$$\mathbb{E}[Y(a) - Y(\pi) \mid A = a, X \in S]$$

$$= \int_{x \in S} \mathbb{E}[Y(a) - Y(\pi) \mid x] p(x \mid A = a, X \in S) dx \qquad \text{Prop. A2}$$

$$= \int_{x \in S} \mathbb{E}[Y(a) - \mathbb{E}[Y \mid x] \mid x] p(x \mid A = a, X \in S) dx \qquad \text{Prop. A1}$$

$$= \int_{x \in S} \mathbb{E}[Y(a) \mid x] p(x \mid A = a, X \in S) dx$$

$$\quad - \int_{x \in S} \mathbb{E}[\mathbb{E}[Y \mid x] \mid x] p(x \mid A = a, X \in S) dx$$

$$= \int_{x \in S} \mathbb{E}[Y \mid X = x, A = a] p(x \mid A = a, X \in S) dx$$

$$\quad - \int_{x \in S} \mathbb{E}[\mathbb{E}[Y \mid x] \mid x] p(x \mid A = a, X \in S) dx \qquad \text{(NUC)}$$

$$= \int_{x \in S} \mathbb{E}[Y \mid X = x, A = a, X \in S] p(x \mid A = a, X \in S) dx$$

$$\quad - \int_{x \in S} \mathbb{E}[\mathbb{E}[Y \mid x] \mid X = x, A = a, X \in S] p(x \mid A = a, X \in S) dx$$

$$= \int_{x \in S} \mathbb{E}[Y - \mathbb{E}[Y \mid x] \mid X = x, A = a, X \in S] p(x \mid A = a, X \in S) dx$$

$$= \mathbb{E}[Y - \mathbb{E}[Y \mid x] \mid A = a, X \in S],$$

where the third-to-last line follows from the fact that the event $\{X = x \wedge A = a\} \iff \{X = x \wedge A = a \wedge X \in S\}$ over the set of $x$ that we are integrating over, and thus does not change the conditional expectation of $Y$. Meanwhile, $\mathbb{E}[Y \mid x]$ is a function of $x$ alone, and so the conditional expectation is equivalent if we condition on additional information $\mathbb{E}[\mathbb{E}[Y \mid x] \mid x] = \mathbb{E}[\mathbb{E}[Y \mid x] \mid X = x, A = a, X \in S]$ as long as this conditional expectation is well-defined, which it will be wherever $p(x \mid A = a, X \in S) > 0$. $\qquad \square$

**Proposition A4.** *The partially maximized population objective from Equation (6) is equivalent (up to a factor of 2) to a weighted sum of the absolute value of each agent's conditional relative agent bias. In other words:*

$$\sum_{a \in \mathcal{A}} \mathbb{P}(A = a \mid X \in S) |\mathbb{E}[Y - \mathbb{E}[Y \mid X] \mid A = a, X \in S]|_{+}$$

$$= \frac{1}{2} \sum_{a \in \mathcal{A}} \mathbb{P}(A = a \mid X \in S) |\mathbb{E}[Y - \mathbb{E}[Y \mid X] \mid A = a, X \in S]|$$

*Proof.*

$$\mathbb{E}[Y - \mathbb{E}[Y \mid X] \mid X \in S]$$

$$= \sum_{a \in \mathcal{A}} \mathbb{P}(A = a \mid X \in S) \mathbb{E}[Y - \mathbb{E}[Y \mid X] \mid A = a, X \in S]$$

$$= \sum_{a \in \mathcal{A}} \mathbb{P}(A = a \mid X \in S) \left( |\mathbb{E}[Y - \mathbb{E}[Y \mid X] \mid A = a, X \in S]|_{+} + |\mathbb{E}[Y - \mathbb{E}[Y \mid X] \mid A = a, X \in S]|_{-} \right)$$

$$= 0$$

where $|x|_- = \min(x, 0)$ denotes the negative part, and where the last line follows from the fact that $\mathbb{E}[Y - \mathbb{E}[Y \mid X] \mid X \in S] = 0$, from the definition of the conditional expectation. This implies that

$$\sum_{a \in \mathcal{A}} \mathbb{P}(A = a \mid X \in S) |\mathbb{E}[Y - \mathbb{E}[Y \mid X] \mid A = a, X \in S]|_+$$

$$= -\sum_{a \in \mathcal{A}} \mathbb{P}(A = a \mid X \in S) |\mathbb{E}[Y - \mathbb{E}[Y \mid X] \mid A = a, X \in S]|_- ,$$

while the weighted sum of the absolute values is given by

$$\sum_{a \in \mathcal{A}} \mathbb{P}(A = a \mid X \in S) |\mathbb{E}[Y - \mathbb{E}[Y \mid X] \mid A = a, X \in S]|$$

$$= \sum_{a \in \mathcal{A}} \mathbb{P}(A = a \mid X \in S) (|\mathbb{E}[Y - \mathbb{E}[Y \mid X] \mid A = a, X \in S]|_+ - |\mathbb{E}[Y - \mathbb{E}[Y \mid X] \mid A = a, X \in S]|_-)$$

$$= 2 \sum_{a \in \mathcal{A}} \mathbb{P}(A = a \mid X \in S) |\mathbb{E}[Y - \mathbb{E}[Y \mid X] \mid A = a, X \in S]|_+$$

where the last line follows from the fact that the weighted sum of the positive parts is equal to the (negative) weighted sum of the negative parts. $\qquad \square$

## A.2  Proof of Theorem 1

**Theorem** (Causal Identification). *Under Assumption 1, $Q(S, G)$ can be identified as*

$$Q(S, G) = \mathbb{E}_S [Cov(Y, G \mid X)] = \mathbb{E}_S [(Y - \mathbb{E}[Y \mid X])G] \qquad (9)$$

*where $\mathbb{E}_S[\cdot] := \mathbb{E}[\cdot \mid X \in S]$, and $Cov(Y, G \mid X)$ is the conditional covariance.*

First, we will prove the following lemma:

**Lemma 1.** *Under Assumption 1, we can write the expected conditional covariance as follows for binary random variables $Y, \mathbf{1}[A = a]$*

$$\mathbb{E}[Cov(Y, \mathbf{1}[A = a] \mid x) \mid X \in S] = \mathbb{P}(A = a \mid X \in S)\mathbb{E}[Y(a) - Y(\pi(x)) \mid A = a, X \in S]$$

*Proof.* The conditional covariance can be written as follows

$$\begin{aligned}
Cov(Y, \mathbf{1}[A = a] \mid x) &= \mathbb{E}[(Y - \mathbb{E}[Y \mid x])\mathbf{1}[A = a] \mid x] \\
&= \mathbb{P}(Y = 1, A = a \mid x) - \mathbb{P}(Y = 1 \mid x)\mathbb{P}(A = a \mid x) \\
&= (\mathbb{E}[Y \mid A = a, x] - \mathbb{E}[Y \mid x])\mathbb{P}(A = a \mid x) \\
&= (\mathbb{E}[Y(a) \mid x] - \mathbb{E}[Y \mid x])\mathbb{P}(A = a \mid x) \\
&= \mathbb{E}[Y(a) - Y \mid x]\mathbb{P}(A = a \mid x)
\end{aligned}$$

where in the penultimate line, we use our causal assumptions to write that

$$\mathbb{E}[Y \mid A = a, x] = \mathbb{E}[Y(a) \mid A = a, x] = \mathbb{E}[Y(a) \mid x],$$

by consistency and no-unmeasured-confounding, respectively. To get the expected conditional covariance, we integrate this over $x \in S$ to arrive at

$$\begin{aligned}
\mathbb{E}[Cov(Y, A \mid x) \mid X \in S] &= \int_{x \in S} \mathbb{E}[Y(a) - Y \mid x]\mathbb{P}(A = a \mid x)p(x \mid X \in S)dx \\
&= \int_{x \in S} \mathbb{E}[Y(a) - Y \mid x]\mathbb{P}(A = a \mid x, X \in S)p(x \mid X \in S)dx \\
&= \int_{x \in S} \mathbb{E}[Y(a) - Y \mid x]\mathbb{P}(A = a, X = x \mid X \in S) \\
&= \mathbb{P}(A = a \mid X \in S)\int_{x \in S} \mathbb{E}[Y(a) - Y \mid x]p(x \mid A = a, X \in S) \\
&= \mathbb{P}(A = a \mid X \in S)\mathbb{E}[Y(a) - \mathbb{E}[Y \mid x]] \mid A = a, X \in S]
\end{aligned}$$

where in the second line, we note that $\mathbb{P}(A = a \mid x) = \mathbb{P}(A = a \mid x, X \in S)$, since the event $\{X = x\} \subset \{X \in S\}$ for any $x \in S$. $\qquad \square$

With this in hand, we can prove Theorem 1 by noting that we can write the function $G$ as $G(A) = \sum_{a:G(a)=1} \mathbf{1}[A = a]$. Using this, we can write that

$$
\begin{aligned}
\mathbb{E}[\text{Cov}(Y, G(A) \mid x) \mid X \in S] &= \mathbb{E}[\text{Cov}(Y, \sum_{a:G(a)=1} \mathbf{1}[A = a] \mid x) \mid X \in S] \\
&= \mathbb{E}[\sum_{a:G(a)=1} \text{Cov}(Y, \mathbf{1}[A = a] \mid x) \mid X \in S] \\
&= \sum_{a:G(a)=1} \mathbb{E}[\text{Cov}(Y, \mathbf{1}[A = a] \mid x) \mid X \in S] \\
&= \sum_{a:G(a)=1} \mathbb{P}(A = a \mid X \in S)\mathbb{E}[Y(a) - Y(\pi(x)) \mid A = a, X \in S]
\end{aligned}
$$

where the second equality follows from linearity of the conditional covariance, the third line follows from linearity of expectation, and the last line follows from Lemma 1

## A.3 Proof of Covariance Identity

In the main text, we claimed that $\mathbb{E}[\text{Cov}(U, V \mid X)] = \mathbb{E}[(U - \mathbb{E}[U \mid X])V]$ for binary $U, V$. This is a known fact, but we give a short a proof here for completeness.

*Proof.* Let $U, V$ be binary random variables, then

$$
\begin{aligned}
\text{Cov}(U, V \mid X) &= \mathbb{E}[(U - \mathbb{E}[U \mid X])(V - \mathbb{E}[V \mid X]) \mid X] \\
&= \mathbb{E}[(U - \mathbb{E}[U \mid X])V \mid X] - \mathbb{E}[(U - \mathbb{E}[U \mid X])\mathbb{E}[V \mid X] \mid X] \\
&= \mathbb{E}[(U - \mathbb{E}[U \mid X])V \mid X].
\end{aligned}
\tag{10}
$$

Here, Eq. (10) follows since for any bounded $f(X)$,

$$
\mathbb{E}[(U - \mathbb{E}[U \mid X])f(X) \mid X] = f(X)\mathbb{E}[U - \mathbb{E}[U \mid X] \mid X] = 0.
$$

The result follows from taking expectation with respect to $X$. $\qquad\square$

## A.4 Proof that $S', G'$ Maximizes Objective 5

*Proof.* For any $S$ and $G$, we have:

$$
\begin{aligned}
Q(S, G) &= \mathbb{E}_S[(Y - \mathbb{E}[Y \mid X])G] \\
&= \mathbb{E}_S\left[(Y - \mathbb{E}[Y \mid X])\left(\sum_a G(a)\mathbf{1}[A = a]\right)\right] \\
&= \sum_a G(a)\mathbb{E}_S[(Y - \mathbb{E}[Y \mid X])\mathbf{1}[A = a]] \\
&= \sum_a G(a)\mathbb{E}_S[Y - \mathbb{E}[Y \mid X] \mid A = a]\mathbb{P}(A = a \mid X \in S).
\end{aligned}
$$

This quantity is maximized by picking $G(a) = 1$ if $\mathbb{E}_S[Y - \mathbb{E}[Y \mid X] \mid A = a] \geq 0$ and $G(a) = 0$ otherwise. If $S$ were disjoint from $S'$, $\mathbb{E}_S[Y - \mathbb{E}[Y \mid X] \mid A = a] = 0$ by Assumption 2. If $S$ intersects $S'$, Assumption 3 implies that $\mathbb{E}_S[Y - \mathbb{E}[Y \mid X] \mid A = a]$ is positive if $G'(a) = 1$ and negative otherwise. In both cases, $G'$ maximizes $Q(S, G)$ for a fixed $S$, i.e. $Q(S, G) \leq Q(S, G')$. By Assumptions 2 and 3, we know that $\text{Cov}(Y, G' \mid X = x) > 0$ for all $x \in S'$, and $\text{Cov}(Y, G' \mid X) = 0$ for all $x \notin S'$. Since $\mathbb{P}(S') = \beta$ and Objective 5 requires $\mathbb{P}(S) \geq \beta$, the optimal choice must be to take $S = S'$. Therefore, $Q(S', G')$ must maximize our objective, as desired, which justifies our writing them as $S^*, G^*$. $\quad\square$

## A.5 Proof of Proposition 1

*Proof.* Notice that

$$\hat{Q}(S, G) = \frac{1}{\sum_{a,j} \mathbf{1}\left[x_{aj} \in S\right]} \sum_{a,j} (y_{aj} - f(x_{aj})) \cdot G(a) \cdot \mathbf{1}\left[x_{aj} \in S\right]$$

$$= \sum_a G(a) \frac{1}{\sum_{a,j} \mathbf{1}\left[x_{aj} \in S\right]} \sum_j (y_{aj} - f(x_{aj})) \cdot \mathbf{1}\left[x_{aj} \in S\right]$$

$$= \sum_a G(a)\hat{Q}(S, \mathbf{1}[A = a]).$$

To maximize this quantity, the optimal choice is $G_S$, where $G_S(a) = 1$ if $\hat{Q}(S, \mathbf{1}[A = a]) \geq 0$, and $G_S(a) = 0$ otherwise. To minimize this quantity, the opposite choice suffices, giving us $G_S^c$. □

## A.6 Proof of Theorem 2

We first present the full version of the Theorem, without the simplifications. Let $\beta'$ denote the fraction of our $N \cdot R$ samples that fall in the region $S^*$. This is the version of the Theorem that we will prove:

**Theorem** (Theorem 2, Formal). *Under the same assumptions as in Section 3.4, as long as $S^*$ is in our hypothesis class $S$ and $R > \frac{2 \ln 2}{\alpha^2 \beta^2 \omega^2}$, the first iteration of Algorithm 1 returns $\hat{S}$ such that, with probability at least $1 - \delta$, $Q(S^*, G^*) - Q(\hat{S}, G^*) \leq \epsilon$, where*

$$\epsilon = \sqrt{\frac{2 \ln(3/\delta)}{\beta' N \cdot R}} + \left(\frac{1}{\beta'} + \frac{1}{\beta}\right)\left(\eta + \sqrt{\frac{3\eta(1 - \eta)}{\delta \cdot N}}\right)$$

$$+ \left|\frac{1}{\hat{\mathbb{P}}(\hat{S})} - \frac{1}{\mathbb{P}(\hat{S})}\right| + \frac{1}{\mathbb{P}(\hat{S})}\left(2\mathcal{R}(S, N \cdot R) + 4\sqrt{\frac{2 \ln(12/\delta)}{N \cdot R}}\right),$$

*where $\mathcal{R}(S, N \cdot R)$ is the Rademacher complexity of $S$, and*

$$\eta = \exp\left(\frac{-R\alpha^2\beta^2\omega^2}{2}\right).$$

For simplicity, we assume that $X$ is a continuous random variable with a well defined density $\mathbb{P}(X = x)$. We first bound the error rate in estimating the grouping on the first iteration.

**Lemma 2.** *Under Assumptions 2, 3, and 4, let $\hat{G}$ be the grouping returned by the first iteration of Algorithm 1. Then for every $a \in A$,*

$$\mathbb{P}[\hat{G}(a) \neq G^*(a)] \leq \eta, \qquad \text{where} \qquad \eta := \exp\left(\frac{-R\alpha^2\beta^2\omega^2}{2}\right). \qquad (11)$$

*with $\alpha$ defined in Assumption 3, $\beta$ given as input to the algorithm, and $\omega$ defined in Assumption 4. As long as $R > \frac{2 \ln 2}{\alpha^2 \beta^2 \omega^2}$, then $\eta < 1/2$*

*Proof.* Choose some agent $a$, and assume that $G^*(a) = 1$, as the argument is symmetric if $G^*(a) = 0$. Define $\hat{Q}_a := \frac{1}{R}\sum_{j=1}^R (y_{aj} - f(x_{aj}))$ as shorthand for the sample average $\hat{Q}(S, \mathbf{1}[A = a])$, where $S = X$ for the first iteration. Then

$$\hat{G}(a) = \mathbf{1}[Q_a \geq 0],$$

and since $G^*(a) = 1$, it suffices to bound the probability that $Q_a < 0$. The expected value of $Q_a$ is given by

$$\mathbb{E}[Q_a] = \mathbb{E}[Y - \mathbb{E}[Y \mid X] \mid A = a]$$

$$= \mathbb{E}\left[(Y - \mathbb{E}[Y \mid X])\mathbf{1}[X \in S^*] + (Y - \mathbb{E}[Y \mid X])\mathbf{1}[X \notin S^*] \mid A = a\right] \qquad (12)$$

$$= \mathbb{E}[(Y - \mathbb{E}[Y \mid X])\mathbf{1}[X \in S^*] \mid A = a]$$

$$= \int_{\tilde{S}} \mathbb{E}[Y - \mathbb{E}[Y \mid X] \mid X = x, A = a]\mathbb{P}(X = x \mid A = a)\, dx \qquad \tilde{S} := \{x : x \in S^* \wedge p(x \mid a) > 0\}$$

$$\geq \int_{\tilde{S}} \alpha\mathbb{P}(X = x \mid A = a)\, dx \qquad (13)$$

$$= \alpha\mathbb{P}(X \in S^* \mid A = a) \qquad (14)$$

$$\geq \alpha\beta\omega, \qquad (15)$$

where the second term in Eq. (12) is zero by Assumption 2; in the inequality on Eq. (13) we have used Assumptions 2 and 3 to lower-bound $\mathbb{E}[Y - \mathbb{E}[Y \mid X] \mid X = x, A = a]$ (recall that $G^*(a) = 1$), and in Eq. (15) we have used Assumption 4 to lower bound $\mathbb{P}(X \in S^* \mid A = a)$ and we have used the definition of $\beta$ as $\mathbb{P}(X \in S^*)$.

Because $f(x) = \mathbb{E}[Y \mid X]$ by assumption, $\hat{Q}_a$ is an average of i.i.d samples $(y_{aj} - f(x_{aj}))$ drawn from $\mathbb{P}(Y - \mathbb{E}[Y \mid X] \mid A = a)$. Since $y_{aj} \in \{0, 1\}$ and $f(x_{i,j}) \in [0, 1]$, each sample is bounded by the interval $[-1, 1]$. By Hoeffding's inequality, the probability of misclassification is bounded as

$$\mathbb{P}(\hat{Q}_a \leq 0) = \mathbb{P}(\mathbb{E}[Q_a] - \hat{Q}_a \geq \mathbb{E}[Q_a]))$$

$$\leq \exp\left(\frac{-R\alpha^2\beta^2\omega^2}{2}\right) =: \eta.$$

since $\mathbb{E}[Q_a] \geq \alpha\beta\omega$. A symmetric argument holds for $G^*(a) = 0$. $\qquad\square$

*Proof of Theorem 2.* After the first iteration, Algorithm 1 returns a grouping $\hat{G}$ and a subset $\hat{S}$. We are interested in the quality of this set $\hat{S}$, relative to the optimal set $S^*$. For an agent $a$, let $\hat{g}_a := \hat{G}(a)$, and let $g_a := G^*(a)$. For notational convenience, let $r_{aj} := y_{aj} - f(x_{aj})$ be the residual in predicting the treatment. We now have $NR$ samples of the form $r_{aj} \cdot \hat{g}_a$. We let $\hat{Q}$ be the empirical expected conditional covariance computed from samples, so that

$$\hat{Q}(S, G^*) = \frac{1}{\sum_{a,j} \mathbf{1}\left[x_{aj} \in S\right]} \sum_{a,j} r_{aj} \cdot g_a \cdot \mathbf{1}\left[x_{aj} \in S\right], \tag{16}$$

and similarly for $\hat{Q}(S, \hat{G})$. Let $\hat{S} = \arg\max_S \hat{Q}(S, \hat{G})$, then we can expand by adding and subtracting identical terms

$$Q(S^*, G^*) - Q(\hat{S}, G^*)$$
$$= [Q(S^*, G^*) - \hat{Q}(S^*, \hat{G})] + [\hat{Q}(S^*, \hat{G}) - \hat{Q}(\hat{S}, \hat{G})] + [\hat{Q}(\hat{S}, \hat{G}) - Q(\hat{S}, G^*)]$$
$$\leq [Q(S^*, G^*) - \hat{Q}(S^*, \hat{G})] + [\hat{Q}(\hat{S}, \hat{G}) - Q(\hat{S}, G^*)] \tag{17}$$
$$= [Q(S^*, G^*) - \hat{Q}(S^*, G^*)] + [\hat{Q}(S^*, G^*) - \hat{Q}(S^*, \hat{G})] +$$
$$[\hat{Q}(\hat{S}, \hat{G}) - \hat{Q}(\hat{S}, G^*)] + [\hat{Q}(\hat{S}, G^*) - Q(\hat{S}, G^*)], \tag{18}$$

where in Eq. (17) we have used the fact that $\hat{S}$ is the maximizer of $\hat{Q}(S, \hat{G})$ in our hypothesis class, and the assumption that $S^*$ is in our hypothesis class. We will bound these terms in order.

**Bounding the first term of Eq.** (18)**:** For any $S$, let $N_S$ be the number of samples $x_{aj} \in S$. Since $\left|r_{aj} \cdot g_a\right| \leq 1$, we have by Hoeffding's inequality that for any $\epsilon_0 > 0$,

$$\mathbb{P}(Q(S^*, G^*) - \hat{Q}(S^*, G^*) > \epsilon_0) \leq \exp\left(-\frac{N_{S^*}\epsilon_0^2}{2}\right) = \exp\left(-\frac{\beta' NR\epsilon_0^2}{2}\right), \tag{19}$$

where we have used the fact that $N_{S^*} = \beta' NR$, by definition of $\beta'$.

**Bounding the second, third terms of Eq.** (18): For any $S$,

$$\hat{Q}(S, G^*) - \hat{Q}(S, \hat{G}) = \frac{1}{N_S} \sum_{a,j} r_{aj} \cdot (g_a - \hat{g}_a) \cdot \mathbf{1}\left[x_{aj} \in S\right]$$

$$\leq \frac{1}{N_S} \sum_{a,j} \left|r_{aj}\right| \cdot |(g_a - \hat{g}_a)| \cdot \mathbf{1}\left[x_{aj} \in S\right]$$

$$\leq \frac{1}{N_S} \sum_{a,j} |g_a - \hat{g}_a| = \frac{RN}{N_S}\left(\frac{1}{N} \sum_a |g_a - \hat{g}_a|\right), \tag{20}$$

where for simplicity, there are $R$ samples per agent by assumption. Note that $|g_a - \hat{g}_a|$ is 1 if $g_a$ is misclassified, and 0 otherwise. Each $\hat{g}_a$ is independently distributed, and by Lemma 2, $p_a := \mathbb{P}(g_a \neq \hat{g}_a) \leq \eta$. Then $\mathbb{E}[\sum_a |g_a - \hat{g}_a|] = \sum_a p_a$, and $\mathbf{Var}[\sum_a |g_a - \hat{g}_a|] = \sum_a p_a(1 - p_a) \leq N\eta(1 - \eta)$, recalling that $\eta < 1/2$. By Chebyshev's inequality

$$\mathbb{P}\left(\left|\sum_a |g_a - \hat{g}_a| - \sum_a p_a\right| > N\epsilon_1\right) \leq \frac{\eta(1 - \eta)}{N\epsilon_1^2}, \tag{21}$$

so that we can bound $\frac{1}{N} \sum_a |g_a - \hat{g}_a|$ with high probability by $\epsilon_1 + \eta$. Note that by the same logic above, the second and third terms of Eq. (18) can be bounded together by observing that their sum is bounded by

$$\left( \frac{RN}{N_{S^*}} + \frac{RN}{N_{\hat{S}}} \right) \left( \frac{1}{N} \sum_a |g_a - \hat{g}_a| \right)$$

**Bounding the fourth term**: Consider $G^* : \mathcal{A} \to \{0, 1\}$ to be fixed, and let $\mathcal{S}$ be our hypothesis class of functions $S : \mathcal{X} \to \{0, 1\}$. For any $S : \mathcal{X} \to \{0, 1\}$, define $f(Z) := (Y - \mathbb{E}[Y \mid X]) \cdot G^*(A) \cdot S(X)$, then (defining $n$ as $NR$, our total number of samples)

$$\hat{Q}(\hat{S}, G^*) = \frac{1}{\hat{\mathbb{P}}(\hat{S})} \left( \frac{1}{n} \sum_{i=1}^{n} f(Z_i) \right), \qquad \text{and} \qquad Q(\hat{S}, G^*) = \frac{1}{\mathbb{P}(\hat{S})} \mathbb{E}[f(Z)],$$

where to deal with the fact that $\mathbb{P}(S) \neq \hat{P}(S)$, we can write that

$$\hat{Q}(\hat{S}, G^*) - Q(\hat{S}, G^*)$$

$$= \left( \frac{1}{\hat{\mathbb{P}}(\hat{S})} - \frac{1}{\mathbb{P}(\hat{S})} \right) \frac{1}{n} \sum_{i=1}^{n} f(Z_i) + \frac{1}{\mathbb{P}(\hat{S})} \left( \frac{1}{n} \sum_{i=1}^{n} f(Z_i) - \mathbb{E}[f(Z)] \right)$$

$$\leq \left| \frac{1}{\hat{\mathbb{P}}(\hat{S})} - \frac{1}{\mathbb{P}(\hat{S})} \right| + \frac{1}{\mathbb{P}(\hat{S})} \left( \frac{1}{n} \sum_{i=1}^{n} f(Z_i) - \mathbb{E}[f(Z)] \right),$$

and we can bound the last term by standard learning theory arguments. In particular, this can be seen as a weighted loss, where $f(Z) = W(Z) \cdot S(Z)$, for $W := (Y - \mathbb{E}[Y \mid X]) \cdot G(A)$. Note that each $S \in \mathcal{S}$ defines some $f \in \mathcal{F}$. In particular, we can write that with probability at least $1 - \delta_1$, for all $f \in \mathcal{F}$, we have

$$\sup_{f \in \mathcal{F}} \left[ \frac{1}{n} \sum_{i=1}^{n} f(Z_i) - \mathbb{E}f(Z) \right] \leq 2\mathcal{R}(\mathcal{F}, n) + 4\sqrt{(2 \ln(4/\delta_1))/n}$$

$$\leq 2\mathcal{R}(\mathcal{S}, n) + 4\sqrt{(2 \ln(4/\delta_1))/n} \tag{22}$$

where we have used the fact that $|f(Z)| \leq 1$, and where $\mathcal{R}$ is the Rademacher complexity (Thm. 26.5.2 of Shalev-Shwartz and Ben-David, 2014). In the second line, we use the fact that $\mathcal{F} = \phi \circ S$, where $\phi(S) = W \cdot S$ is a 1-Lipschitz function, since $|W| \leq 1$. By the contraction lemma (Lemma 26.9 of Shalev-Shwartz and Ben-David, 2014), we have it that $\mathcal{R}(\mathcal{F}, n) \leq \mathcal{R}(\mathcal{S}, n)$.

**Combining bounds**: By the union bound, the bounds in Eq. (19), Eq. (21) and Eq. (22) hold with probability at least $1 - \delta$, where

$$\delta = \exp\left( -\frac{\beta' N R \epsilon_0^2}{2} \right) + \frac{\eta(1 - \eta)}{N \epsilon_1^2} + \delta_1. \tag{23}$$

Hence, with probability at least $1 - \delta$, we can bound Eq. (18) by

$$Q(S^*, G^*) - Q(\hat{S}, G^*) \leq \epsilon_0 + \left( \frac{1}{\beta'} + \frac{1}{\beta} \right)(\eta + \epsilon_1) + \left| \frac{1}{\hat{\mathbb{P}}(\hat{S})} - \frac{1}{\mathbb{P}(\hat{S})} \right|$$

$$+ (1/\mathbb{P}(\hat{S})) \left( 2\mathcal{R}(\mathcal{S}, NR) + 4\sqrt{(2 \ln(4/\delta_1))/NR} \right),$$

where we have used $NR/N_{S^*} = 1/\beta'$ and $NR/N_{\hat{S}} = 1/\beta$, by definition of $\beta'$ and because $\hat{S}$ is chosen to be a $\beta$-fraction of the dataset. Finally, we simplify by setting the three terms in Eq. 23 to be equal, and expressing $\epsilon_0$ and $\epsilon_1$ in terms of $\delta$:

$$\epsilon_0 = \sqrt{\frac{2 \ln(3/\delta)}{\beta' NR}}, \quad \epsilon_1 = \sqrt{\frac{3\eta(1 - \eta)}{\delta N}}, \quad \delta_1 = \delta/3$$

from which the desired result follows. □

# B  Semi-Synthetic Experiments: Additional Details

## B.1  Semi-Synthetic Setup Details

**Additional Dataset Details**: The Stanford study on human predictions of recidivism only included participants who passed both attention checks in the assessment. Participants were provided information about the baseline recidivism rates and charge. We use the responses from participants who were not given feedback so that decisions from the same participant are independent and identically distributed. The participants provide probabilistic predictions, which we convert into binary decisions by thresholding at 50%. By construction of the survey, no participant can predict exactly 50%.

**Semi-Synthetic Policy Generation (Details)**: For both alternative policies (as logistic regression models), we add a new binary feature corresponding to the binary region indicator (1 if the sample falls in the region, 0 otherwise). The weight of this new coefficient is 1.5 for the alternative policy and 0 for the base policy. We give additional details on the differences between the base and alternative policies below:

- *Drug Possession*: On average, the base and alternative policies predict recidivism 56% and 61% of the time, respectively, outside the subset of drug-possession charges. Within the subset, they predict recidivism 60% and 87% of the time, respectively.

- *Misdemeanor and Age at Most 35*: On average, the base and alternative policies predict recidivism 56% and 62% of the time, respectively, outside the subset of misdemeanor charges for individuals with age at most 35. Within the subset, the averages are 48% and 79% respectively.

## B.2  Baseline Details

In this section, we describe implementation details for the two baselines discussed in section 4.1, two additional baselines we created, limitations of the baselines, and additional evaluation on the semi-synthetic set-ups.

**Direct Model:** The agents are included in a one-hot encoding, with the last agent dropped to prevent co-linearity. The agents are partitioned into two equal-size groups based on their logistic regression coefficients. Logistic regressions (LR), ridge regressions (RR), decision trees (DT), and random forests (RF) are via scikit-learn (Pedregosa et al., 2011). For logistic regressions, we tune the L2 regularization constant ($C$) among 10, 1, 0.1, 0.01, 0.001, 0.0001, and 0.00001. For ridge regressions, we tune the L2 regularization constant ($\alpha$) among 0.01, 0.1, 1, 10, and 100. For decision trees and random forests, we tune the minimum samples per leaf among 10, 25, and 100. For random forests, we also tune the number of trees among 10, 25, and 100.

**TARNet**: Our optimizer performs stochastic gradient descent with zero momentum on the binary cross entropy loss function from PyTorch (Paszke et al., 2019). We tune the learning rate among 0.0001, 0.001, and 0.01 for the drug possession set-up and among 0.0001, 0.0005, 0.001, 0.005, and 0.01 for the misdemeanor set-up. We tune the number of units in each of the layers among 10 and 20 for the drug possession set-up and among 10, 20, and 30 for the misdemeanor set-up. Finally, we tune dropout among 0.05 and 0.25. We train the model for 200 epochs and use the epoch that has the lowest validation loss.

**U-learners:** A standard U-learner (Nie and Wager, 2017) is designed to estimate the effect of a binary treatment by predicting treatment from features $T = f(X)$, predicting outcome from features $Y = g(X)$, and then predicting the ratio between the residuals of the previous two models from features $(Y - \hat{g}(X)) / (T - \hat{f}(X)) = h(X)$. To adapt this model to multiple discrete treatments, we fit $f_{ij}$ for every pair of agents $A_i$ and $A_j$ using only the samples from those two agents. This means there is a separate $h_{ij}$ for each pair of agents. The amount of variation for a sample $x$ is defined as $\sum_{i,j \in \mathcal{A}} |\hat{h}_{ij}(x)|$. As in the direct models, a model is fit to predict this quantity, and the top $\beta$ quantile of these predictions is used to identify region membership. To create $\hat{G}$, we first identify a pair of agents that are most dissimilar using the sum of absolute values of the difference between that pair across all samples within the region. All the other agents are added to the partition based on which provider in the starting pair they are closer to.

**Causal Forest Adaptations:** A causal forest (Wager and Athey, 2018) resembles a random forest with the splits chosen instead to maximize the difference in the effect of the treatment on outcome. A causal forest can only predict the difference between two treatments. The naïve adaptation of fitting $O\left(|\mathcal{A}|^2\right)$ causal forests between each pair of agents becomes computationally infeasible when $|\mathcal{A}|$ is large. Instead, we use the following heuristic to learn a partition with causal forests: Initialize the groupings $G_0$ and $G_1$ with a few agents based on the U-learner predictions. We start with the most dissimilar pair as defined above. Then, if a provider is much closer to one of the two starting providers (defined as the sum of absolute differences being at most a tenth of the sum for the other pair), add that provider to the group with the closer provider. To add agent $a$, compute a causal forest for the difference in treatment choice between $G_0 \cup a$ and $G_1$ and the difference between $G_0$ and $G_1 \cup a$. We use the EconML implementation of causal forests (Research, 2019). For hyperparameter tuning, predictions $U_{a_0,a_1}(X)$ from the U-learner are used as "oracle" predictions. The "oracle" predictions between groups $G_0, G_1$ are defined as $\sum_{a_0 \in G_0, a_1 \in G_1} U_{a_0,a_1}(X)$. We compare each pair of causal forests and add $a$ to $G_0$ if the former has a larger sum of the absolute difference across all samples. Otherwise, add $a$ to $G_1$. Repeat until all agents have been added to a group. A region model is applied to the predictions from the final causal forest.

**Hyperparameter Tuning:** For both the baselines and our algorithm, we tune the hyperparameters for each piece of the model separately, e.g. each decision tree piece of the model is tuned separately. The data is split into train, validation, and test 60/20/20 stratified by provider, guaranteeing at least 1 sample per provider in each of the sets. The same validation set is used for all steps in a single model. After the best hyperparameter is selected, the model is retrained on both the training and validation data.

**Baseline Limitations:** A limitation of the TARNet baseline is that it does not learn a partition of providers. A limitation of the other baselines except for the U-learner is that the provider partition is based on all samples, not just samples within the region.

**Additional Baseline Results:** Other metrics we consider are the precision and recall of the region and the accuracy of the partition. To compute accuracy, we compare the learned groups 0 and 1 with the true groups 0 and 1, respectively, and with the true groups 1 and 0, respectively, and take the higher of the two. Figure 3 shows that the ridge regression model we highlight in the main paper fits the region better than the other two models for large numbers of agents for the drug possession set-up. On the other hand, for the misdemeanor under age 35 set-up, Figure 4 shows that the random forest model fits the region better. TARNet is the best-performing baseline for the drug possession set-up, while the direct model is the best-performing baseline for the misdemeanor set-up. Our model outperforms both in all cases. The partition is difficult for any model to learn in both set-ups when the number of agents increases. Despite this challenge, our algorithm is still able to recover the region reasonably well.

## B.3 Convergence Analysis

We derive a generalization bound on the performance of the iterative algorithm after one iteration in Section 3.4. Here we examine empirical performance across additional iterations. To do so, we compute the AUC of the region in each iteration and how many iterations the algorithm needs to converge. Note that because AUC is not the objective that we are optimizing, there are no guarantees that it increases monotonically across iterations. As seen in Figures 5-6, in both semi-synthetic set-ups, for small numbers of agents, convergence is almost immediate. For larger numbers of agents, the algorithm almost always converges. The exceptions are where the algorithm enters what appears to be a cycle. This problem can likely be addressed by re-initializing the model if the algorithm is stuck in a cycle.

## B.4 Robustness Analysis

We examine robustness of the model to violations of the assumption that there are 2 agent groups. For instance, it may be more realistic for agents to have a wide spectrum of preferences. We represent this in our semi-synthetic set-up by varying the coefficient on the region indicator variable described in Appendix B.1 in equally spaced steps between -1.5 and 1.5. The number of agents is held constant at 40. The region and the policy outside the region are the same as before. This corresponds to varying the average predicted probabilities of recidivism within the subset in the groups between around 30%

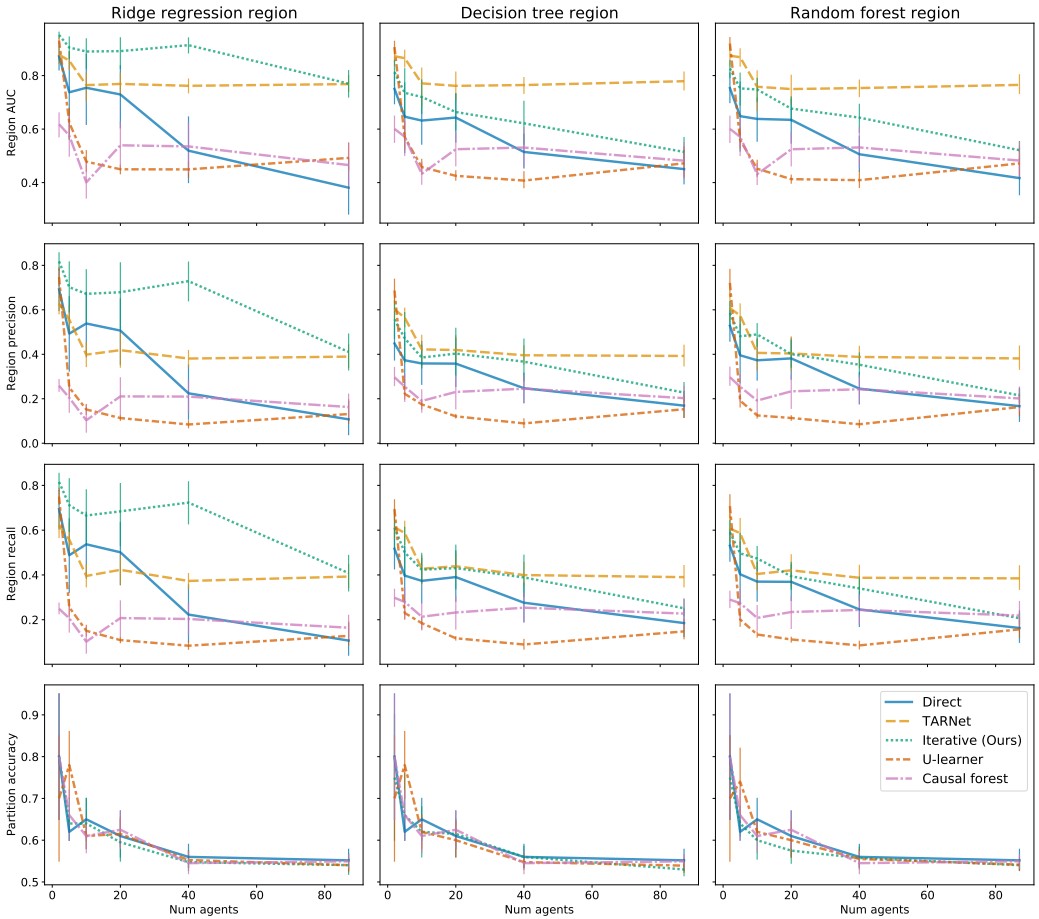

Figure 3: Comparison to baseline approaches for drug possession semi-synthetic set-up. Uncertainty bands represent 95% intervals for the mean derived via bootstrapping computed using seaborn (Waskom, 2021). Our method with a ridge regression region model is the best at identifying the region across all combinations of outcome and region models.

and 86% for the drug possession set-up and between around 17% and 79% for the misdemeanor under age 35 set-up. We compare our method with the direct baseline with all region models. As seen in Figure 7, the ridge regression model is most robust to this violation in the drug possession set-up. In fact, performance actually improves when the number of agent groups increases. We hypothesize this may occur for two reasons: 1. The ridge regression model parametrizes the region best, as that was also the globally best model in the 2-group specification. 2. The agents can still be divided into two groups based on above and below average preference, so this violation of the 2-group assumption is relatively simple. Figure 8 shows that all 3 region models are somewhat robust to increasing numbers of groups for both the iterative and direct baseline in the misdemeanor under 35 set-up.

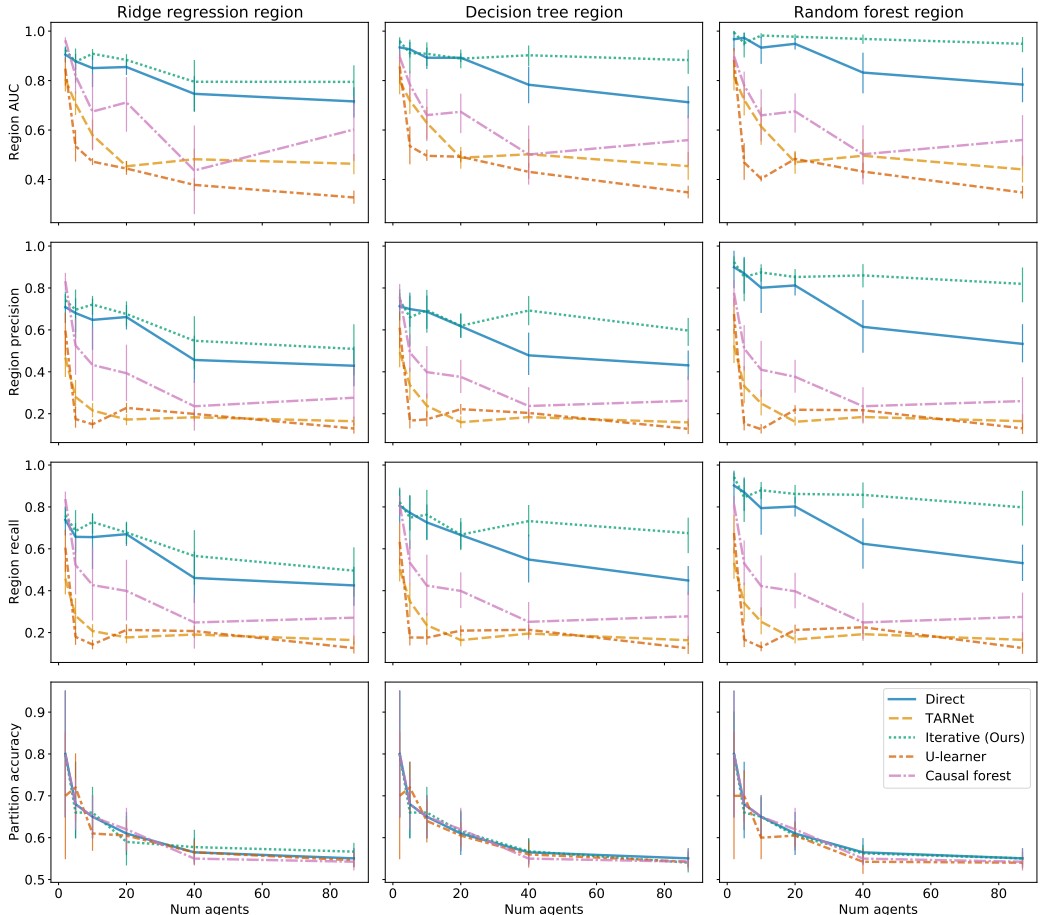

Figure 4: Comparison to baseline approaches for misdemeanor semi-synthetic set-up. Uncertainty bands represent 95% intervals for the mean derived via bootstrapping computed using seaborn (Waskom, 2021). Our method with a random forest region model is the best at identifying the region across all combinations of outcome and region models.

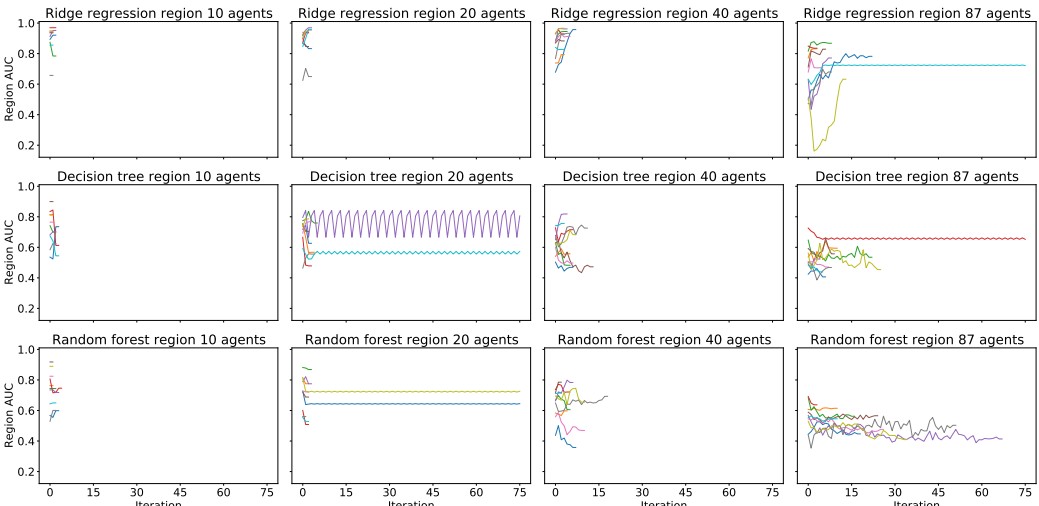

Figure 5: Convergence of iterative algorithm in drug possession semi-synthetic set-up. Each of the 10 lines in each plot represents a dataset generated from a different random seed. Region AUC is computed after the partition is updated. Algorithm terminates when partition does not change. Although the iterative algorithm was run for up to 100 iterations, the number of iterations plotted was truncated to show all terminations. 2 and 5 agents are omitted because they closely resemble the plots for 10 agents.

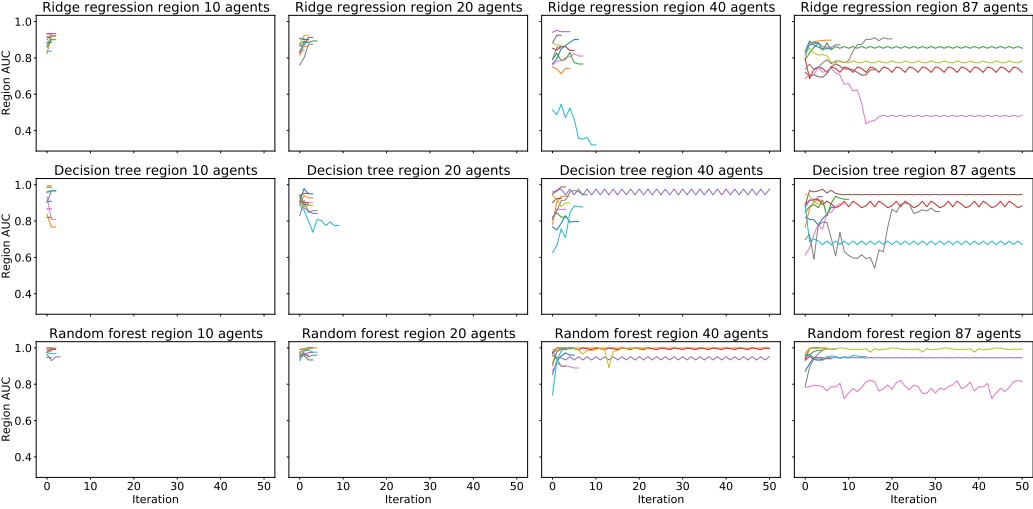

Figure 6: Convergence of iterative algorithm in misdemeanor semi-synthetic set-up. See Figure 5 for description.

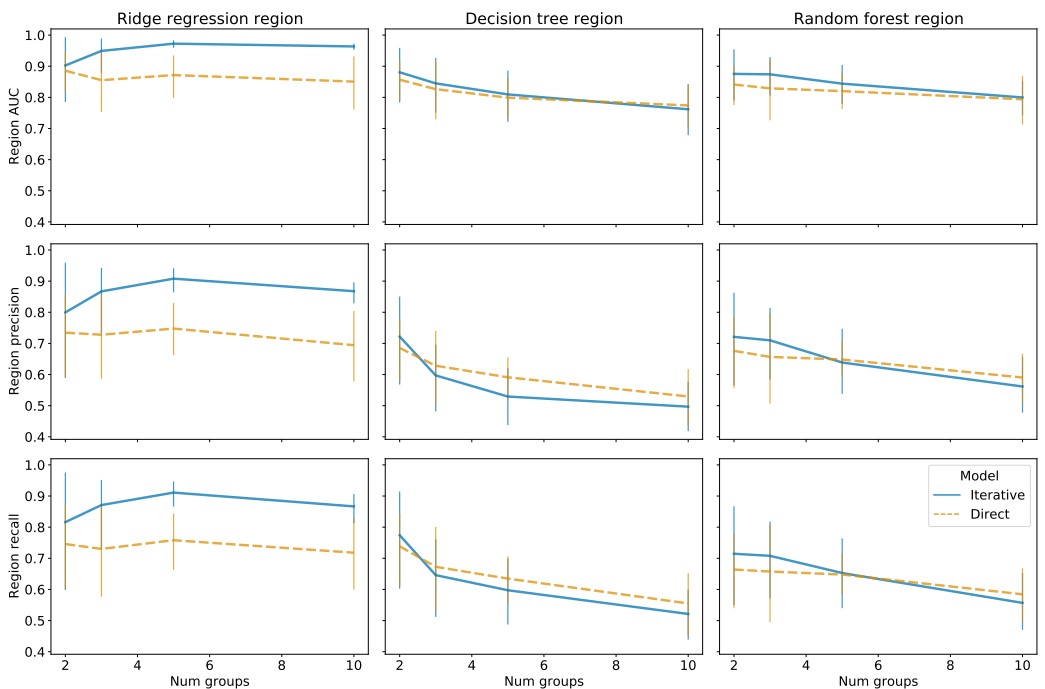

Figure 7: Robustness to the assumption that there are 2 agent groups in the drug possession semi-synthetic set-up. The 40 agents in each set-up are roughly equally divided into 2, 3, 5, and 10 groups. Uncertainty bands represent 95% intervals for the mean derived via bootstrapping computed using seaborn (Waskom, 2021).

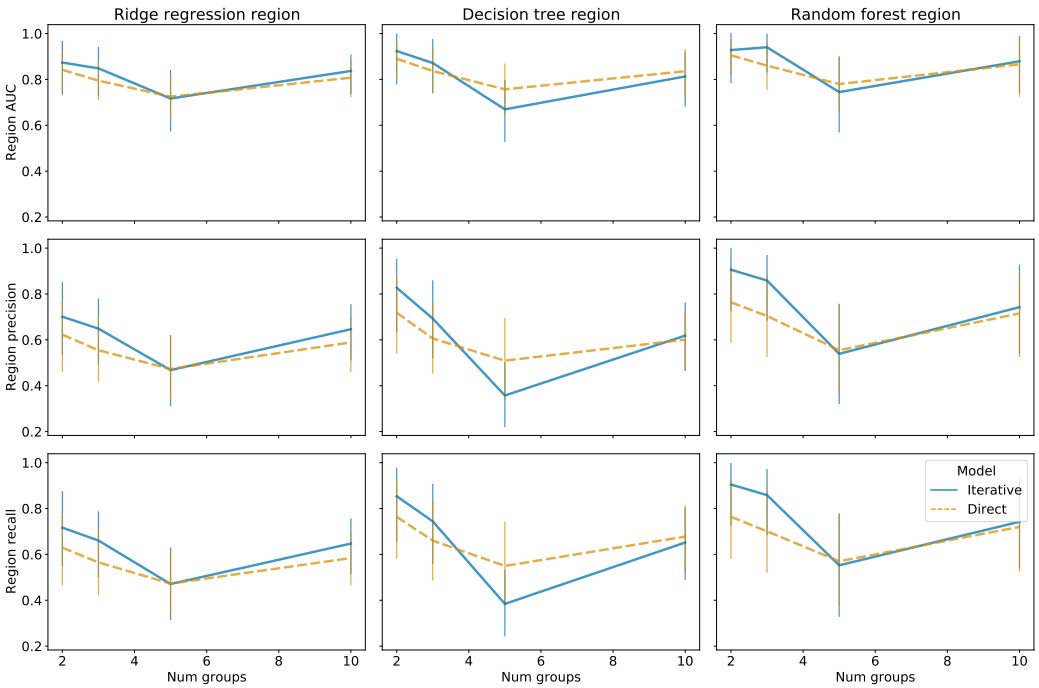

Figure 8: Robustness to the assumption that there are 2 agent groups in the misdemeanor under age 35 semi-synthetic set-up. See Figure 7 for description.

# C  Diabetes Experiment: Additional Details

**Data Details, Train/Validation/Test Split**: The data set of de-identified health insurance claims was provided by a large insurance company. That company obtained the relevant consent from individuals to use their data, and gave us permission to use the data for research purposes. A waiver of informed consent was obtained in compliance of HIPAA.

We require at least 3 years of observation before the first diabetes treatment to ensure that the observed treatment is indeed first-line. We also require at least one diagnosis code of diabetes mellitus and at least one A1C measurement at least 6.5 prior to first-line treatment. Patients who had at least one diagnosis code related to type 1 diabetes mellitus prior to first-line treatment or gestational diabetes (pregnancy, neonatal diabetes, or diabetes of the young) in the 1 year prior to first-line treatment are excluded. The exclusion of gestational diabetes is important to prevent confounding since those patients may see specialized providers and receive different treatments. Patients who received more than one first-line treatment or any treatment besides metformin, DPP-4 inhibitors, or sulfonylureas are also excluded.

We only include agents with at least 4 samples. For each agent, 37.5% of samples are placed in the training set, 12.5% in validation, and 50% in test to ensure the test set is sufficiently large for computing the partition in $L(\hat{S})$ on the test set. We require at least one sample per agent in each of the training and validation sets and two in the test set. Treatment date is converted to seconds, and all features are normalized to have mean 0 and standard deviation 1.

**Selecting the outcome model**: For the outcome model, we consider three hypothesis classes: logistic regressions, decision trees, and random forests. The models are tuned as described in Appendix B.2. We select a random forest for three reasons: 1. Random forests have the highest AUC on the validation set, as shown in Table 2. 2. Random forests are better calibrated in the discovered region, as shown in Table 3. 3. The partial dependence plots in Figure 9 show reasonable relations between each feature and the outcome. These results are reported for the fold that was selected based on significance and calibration statistics.

Table 2: Validation AUCs for outcome model for diabetes experiment.

| Model | Test AUC |
|---|---|
| Logistic regression | 0.6957 |
| Decision tree | 0.7351 |
| Random forest | 0.8283 |

Table 3: Calibration of outcome model on regions selected with each outcome model for diabetes experiment. Comparison of average true and predicted outcomes in region among training samples.

| Model | True average | Predicted average |
|---|---|---|
| Logistic regression | 0.2485 | 0.1839 |
| Decision tree | 0.2023 | 0.1688 |
| Random forest | 0.2343 | 0.2226 |

**Visualizing the Region**: To describe the region in an interpretable way, we use a decision tree for the region model $h(x)$ described in Algorithm 1 with $\beta = 0.25$ and a maximum of 5 iterations. The minimum number of samples per leaf for the decision tree region model is tuned among larger choices (50, 100, 200) for a more interpretable region. Our algorithm outputs a decision tree $h(x)$, shown in Figure 10a and a threshold of $b = 0.0741$ in the training $Q(S, G)$ values. The region $S = \{x \in \mathcal{X}; h(x) \geq b\}$ consists of the two nodes indicated in red. To verify the generalizability of this result, we use a held-out test set to compute the metric $L(\hat{S})$ In the test set, this metric is also greater on these two nodes than on all other nodes in the decision tree.

We zoom into the first region in Figure 10b. We only include decisions made by providers with at least two samples in that node. Providers in group $G = 0$ prefer to initiate treatment with metformin, while providers in group $G = 1$ prefer to initiate with sitagliptin, glipizide, glimepiride, or glyburide. This

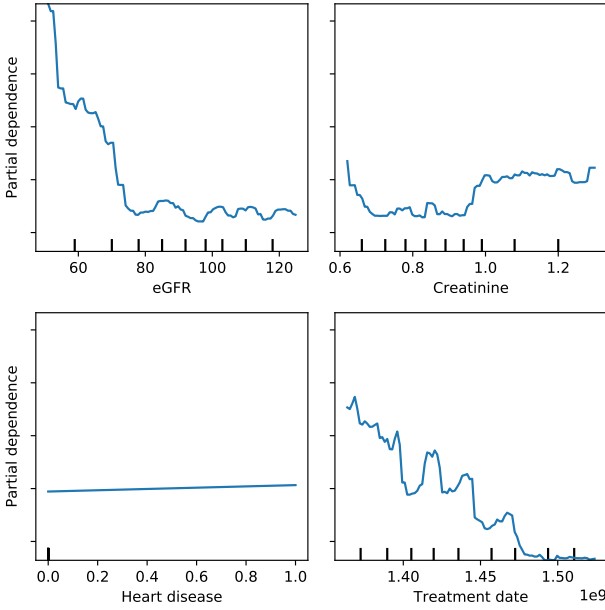

Figure 9: Partial dependence plot of random forest outcome model in diabetes experiment

group-specific bias (blue vs. orange lines) generally holds across the range of GFR and creatinine values, indicating that this preference is not explained by the patient's features.

**Assessing Stability**: We also measure the stability of the region and grouping identified by our method by performing cross-validation. We split the training and validation data into 4 equally sized portions and assign 1 portion as the validation set. The only overlap between the 4 validation folds are samples that belong to agents with fewer than 4 observations in the training and validation set to ensure that all validation folds contain at least 1 sample per agent. (1) A region is stable if points that are selected for the region from most training folds are also selected when they are in the validation fold. If we look at the points that are in 1 validation fold, among the 461 points that are in the region for 2 to 3 training folds, 309 are also selected when they belong to the validation fold. (2) We also examine the consistency of the test region. With an average test region size of 553.75 points, 350.75 points are in the test region for at least 3 folds (each point in only 3 folds contributes weight 0.75). (3) We assess stability of the grouping by examining whether pairs of providers are consistently on the same or opposite sides of the grouping. Among the 12,181 pairs of providers that have at least 1 training, validation, or test sample in the region in at least 3 of the folds, 10,251 pairs have the same relationship in at least 3 of the folds. All 3 of these statistics suggest our algorithm arrives at similar regions and groupings regardless of how the training and validation samples are split.

## D   Additional Real-Data Experiment: First-Line Parkinson's Treatment

**Context and Data**: The Parkinson's Progression Markers Initiative (PPMI) is an observational study that follows Parkinson's patients starting within 2 years of diagnosis in their *de novo* cohort (Marek et al., 2011). The study collected data across the US, Europe, Israel, and Australia between 2010 and 2018. We examine decisions between the two most common first-line treatments, levodopa and rasagiline. Clinical trials are interested in assessing the effects of these treatments (Group et al., 2014). For context, we include age, disease duration, and a motor assessment (the total from part II and III of the Movement Disorder Society Unified Parkinson's Disease Rating Scale (MDS-UPDRS)) (Goetz et al., 2007). The features are normalized to have mean 0 and standard deviation 1. The cohort consists of 260 patients at 23 study sites, which we use as "agents" $A$. Note that while treatment decisions are not made at study sites, these sites capture rough geographic locations across which there may be heterogeneity in treatment. We fix the outcome model to be a decision tree and the region size $\beta = 0.25$.

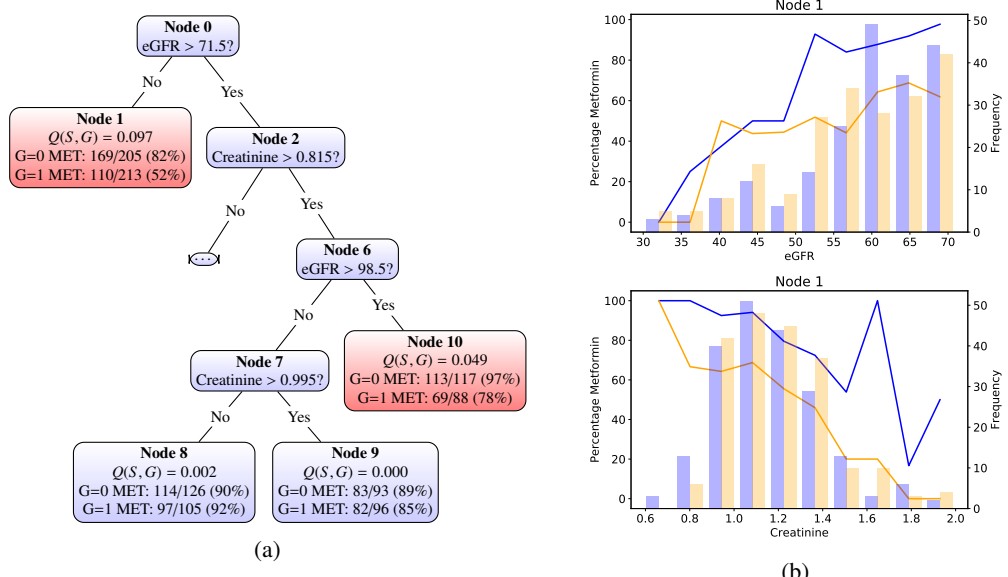

(a)

(b)

Figure 10: (a) Decision tree identifying regions of disagreement for first-line treatment decisions in diabetes, indicated in red. MET: Metformin. G=0 denotes the group found to prefer metformin as an initial treatment, and G=1 indicates the group with the opposing preference. Numbers at leaves computed on a held-out test set. $Q(S, G)$ values are the $L\left(\hat{S}\right)$ metric computed on the test set; number of samples are computed using only providers with at least two samples in the region of heterogeneity. (b) Variation in Node 1 of the tree in Figure 10a, one of the regions of disagreement in the diabetes dataset. The colors denote group membership: blue is $G = 0$, and orange is $G = 1$. The lines indicate the proportion of decisions in each bin where the agent prescribed metformin. The gap between the lines illustrates that the group-specific bias towards metformin generally holds across patient features. Only agents with at least two samples in the region are included. The histograms show the total number of samples in each bin.

**Interpretation of Results**: The decision tree in Figure 11 shows that the selected region includes patients who are either above age 70 or both above age 62 and diagnosed at least around 1.5 years prior to first-line treatment. The region in the test set includes 13 patients in group 0 and 40 patients in group 1. Patients in the region in the test set are older on average (72.3 versus 64.3 for entire test set), have had Parkinson's longer (1.9 years versus 1.5 years) and have higher MDS-UPDRS part II + III scores (41.7 vs 37.5). Empirically, when stratified on age or MDS-UPDRS, patients in the top 33rd percentile (67 for age, 29 for MDS-UPDRS) are more likely to be given levodopa, while patients in the lower 67th are more likely to receive rasagiline.

This region likely captures where treatment switches from rasagiline to levodopa. Our results align with clinical guidelines that levodopa may be better for patients with more motor and cognitive impairment due to fewer side effects (Muzerengi and Clarke, 2015), as these patients also tend to be older and have had the disease longer.

**Assessing Significance**: In Table 4, we assess whether the region $S$ identifies variation in held-out data better than a randomly selected region. As in the diabetes experiment, we observe that the test statistics is close to the training statistic and more than 2 standard deviations from the average test statistic for random regions of the same size, suggesting the discovered region of heterogeneity generalizes beyond the training set.

**PPMI Disclaimer**: Data used in the preparation of this article were obtained from the Parkinson's Progression Markers Initiative (PPMI) database (www.ppmi-info.org/data). For up-to-date information on the study, visit www.ppmi-info.org. PPMI – a public-private partnership – is funded by the Michael J. Fox Foundation for Parkinson's Research and funding partners, including abbvie, AcureX therapeutics, Allergan, Aligning Science Across Parkinson's, Avid Radiopharmaceuticals, Bial Biotech, Biogen, BioLegend, Bristol Myers Squibb, Calico, Celgene, Dacapo brainscience,

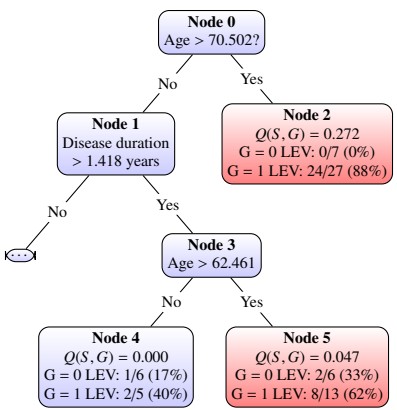

Figure 11: Decision tree identifying regions of disagreement for first-line treatment decisions in Parkinson's, indicated in red. LEV: Levodopa. G=1 denotes the group found to prefer levodopa as an initial treatment, and G=0 indicates the group with the opposing preference. See Figure 10a for explanation.

Table 4: Objective values $L(S)$ for the learned region on the training and test datasets, along with the distribution of values for randomly generated regions $S_{\text{rand}}$ given as mean (standard deviation).

| Metric | Subset | Value |
|---|---|---|
| $L(\hat{S})$ | Train | 0.2743 |
| $L(\hat{S})$ | Test | 0.2170 |
| $L(S_{\text{rand}})$ | Test | 0.1200 (0.0222) |

Denali, Edmond J. Safra Philanthropic Foundation, 4D Pharma PLC, GE Healthcare, Genentech, GlaxoSmithKline, Golub Capital, Handl Therapeutics, insitro, Janssen Neuroscience, Lilly, Lundbeck, Merck, Meso Scale Discovery, Neurocrine biosciences, Pfizer, Piramal, Prevail Therapeutics, Roche, Sanofi Genzyme, Servier, Takeda, Teva, ucb, verily, Voyager Therapeutics, and Yumanity Therapeutics.