# OpenReview forum: "Finding Regions of Heterogeneity in Decision-Making via Expected Conditional Covariance"
_NeurIPS.cc/2021/Conference — NeurIPS 2021 Poster_

### Official Review · Reviewer_Y7AH · 2021-07-13

**Rating:** 7
**Confidence:** 3

**Summary:**

The authors are concerned with identifying regions in a covariate space. A region is characterised by having high variation in (potential) treatment outcome, as compared to the expected (potential) outcome. A method such as this could be used to identify variation of practice in the legal or medical domain (as indicated by the authors).

This is a relevant topic which the authors rightfully relate to the causal literature. Specifically, the authors chose the potential outcomes model, where the treatment is now a clinician or judge; the outcome is binary (amounting to a decision, e.g. bail or no bail); and the condition is an individual. Contrasting potential outcomes, the authors are interested in learning subsets of X, rather than Y(a).

**Limitations And Societal Impact:**

The work presented in this paper taps into an active area of research (subgroup analysis of HTE). The biggest limitation, in my opinion, is (largely) not acknowledging this existing brand of literature. Furthermore, I believe the authors should benchmark against (at least some of) these existing methods (cited above).

**Main Review:**

I found the paper clear and well written, the topic relevant, and the presented method interesting. However, there are a few concerns which I have listed below.

### Major
* On overlap. In Assumption~1 (line 68), the authors do not assume overlap (0 < p(A | X =x) < 1 for all x). They motivate this in the remark starting at line 104. The motivation seems flawed to me; line 108 "If judge a makes bail decisions primarily for violent crimes, but some other judges never see violent crimes, then we cannot hope to compare Y(a) to the decisions of those alternative judges.", this is correct. However, what if we want to know the potential outcome of placing a violent crime case to those alternative judges? The reason we make the overlap assumption is such that we can ask the model for an estimate of these edge-cases. By using a propensity (such as the authors suggest on line 111), the contribution of those alternative judges is minimal, but if we wish to make an estimate for these judges, we do want some data which we can consider. If those propensities were 0, we would have a biased estimate.
* There exists literature on partitioning for heterogeneous treatment effects which remains undiscussed [1, 2, 3, 4, 5]. This literature contains important discussion, as well as important benchmarks for the authors' experiments.

### Minor
* line 94, "[...] if the agent-specific bias is non-negative on S.", I'm guessing this should be non-zero? If the bias is negative, then there is still bias, correct?

[1] HS Lee et al. "Robust Recursive Partitioning for Heterogeneous Treatment Effects with Uncertainty Quantification" NeurIPS 2020

[2] K Imai et al. "Estimating treatment effect heterogeneity in randomized program evaluation." Annals of Applied Statistics 2013

[3] JC Foster et al. "Subgroup identification from ran- domized clinical trial data." Statistics in Medicine 2011

[4] LL Doove et al. "A comparison of five recursive partitioning methods to find person subgroups involved in meaningful treatment– subgroup interactions." Advances in Data Analysis and Classification 2014

[5] S Athey & G Imbens "Recursive partitioning for heterogeneous causal effects." PNAS 2016

**Time Spent Reviewing:**

3h

---

> ### Author Response · Authors · 2021-08-10
> **Response to Reviewer Y7AH**
>
> Thank you for the detailed review.  We provide a few clarifications below.
>
> **Differences from partitioning methods**: The reviewer highlights relevant literature about partitioning the sample space into subgroups for heterogeneous treatment effect estimation and requests that we benchmark against these methods.
>
> First, we would like to highlight a fundamental difference: These methods assume a binary treatment, so that the treatment effect at each point is a single number. In contrast, our problem involves estimating the differences between the policies of $A$ agents (so there are $A(A-1)/2$ pairs of agents, and as many “treatment effects”).  Since we have a large number of agents in our real-world experiment ($A = 458$--see line 280), it is not obvious how to apply these methods out-of-the-box to our problem.  Nonetheless, we do attempt to adapt these types of partitioning methods (such as causal forests [1]) as well as additional HTE methods (such as U-learner [2]) as additional baselines (see lines 695-721 in the supplement). In light of this discussion, we will bring these points and our additional baselines into the main text if space permits.
>
> **Overlap assumption**:  In an ideal world, we might prefer that overlap holds, as this (along with our other assumptions) would enable us to estimate causal effects like E[Y(a) - Y(a’)] for any pair of agents.
>
> However, our main point is this:  Even when overlap fails (in the setting of many agents), we can still estimate (different) causal effects that may be of interest.  For instance, consider the following example, from our discussion with reviewer DZe8 (see point #1) - suppose that $X$ is binary, and that when $X = 0$, agents $a\_1, a\_2$ see cases with equal probability, and when $X = 1$, agents $a\_3, a\_4$ see cases with equal probability.  In this setting, it is impossible to compare the decisions of agents $a\_1$ and $a\_3$, because they never see the same value of $X$.  Nonetheless, we can still identify disagreement between $a\_1$ and $a\_2$, and between $a\_3$ and $a\_4$, which is what our objective captures.
>
> **Non-negative bias**: Finally, regarding line 94, “non-negative bias” is written as intended. We define the first group ($G(a) = 1$) as agents with non-negative bias, with the rest of the agents belonging to the second group ($G(a) = 0$).  With this assignment of $G$, the resulting quantity $Q(S, G)$ is the same (up to a constant) as taking the weighted sum of the absolute values of the bias for each agent.  See our discussion with reviewer DZe8 for more details (see our response to points 2 & 3).
>
> [1] Stefan Wager and Susan Athey. 2018. Estimation and inference of heterogeneous treatment effects 476 using random forests. J. Amer. Statist. Assoc. 113, 523 (2018), 1228–1242.
>
> [2] Xinkun Nie and Stefan Wager. 2017. Quasi-oracle estimation of heterogeneous treatment effects. 434 arXiv preprint arXiv:1712.04912 (2017).

---

> > ### Comment · Reviewer_Y7AH · 2021-08-30
> > **Dear authors**
> >
> > Dear author(s),
> >
> > _My apologies for not responding sooner._
> >
> > I find your response to mostly clarify my comments and will improve my score from 6->7.

---

### Official Review · Reviewer_DZe8 · 2021-07-13

**Rating:** 6
**Confidence:** 5

**Summary:**

In this paper, the authors develop an algorithm for identifying cases (e.g., defendants or patients) on which decision makers (e.g., judges and doctors) "disagree" on the decision.

## Identification
The authors consider a setting in which cases are summarized by features $X$. Each case is associated with a decision maker $A$ and a binary decision $Y$. In the pretrial release setting, each case is a defendant, the features $X$ are observable characteristics of the defendant (such as demographics or charge information), the decision maker $A$ is the assigned judge and the binary decision $Y$ is to either release or detain the defendant. The authors define the potential decision Y(a), which describes the decision that would be made by decision maker $a$.

The authors propose a summary measure of agent disagreement using average contrasts of potential decisions across agents given sets of the observable features. In particular, the authors focus on the "conditional relative agent bias" as
$$ \mathbb{E}[Y(a) - Y(\pi(x) | A = a, X \in S], $$
where $Y(\pi(x))$ denotes the expected decision under the assignment mechanism of decision makers to cases. The authors key identification assumption is that decision makers are as-if randomly assigned to cases conditional on the features (i.e., $A \perp Y(a) \mid X$). This measures the extent to which decision maker a's average decisions over $X \in S$ differs from the average decision that is made under the assignment mechanism. In other words, this measure of disagreement depends on the assignment mechanism of agents.

The conditional relative agent bias measures the extent to which a single agent a's decision disagrees relative to the average decision over $X \in S$. The authors then propose aggregating this measure across agents. For some grouping of agents G, the authors define the estimand
$$ Q(S, G) = \sum_{a : G(a) = 1} P(A = a \mid X \in S) \mathbb{E}[Y(a) - Y(\pi(x) | A = a, X \in S]. $$
The authors main identification result (Theorem 1) is to show that if agents are as-if randomly assigned to cases, then this estimand can be identified as the conditional covariance between the observed decision and the agent assignment
$$ Q(S, G) = \mathbb{E}[Cov(Y, G \mid X ) \mid X \in S]. $$

## Estimating Regions of Heterogeneity.
Using this identification result, the authors goal is to search over regions of the feature space $S$ and groupings over agents $G$ to find the combination that maximizes this proposed measure of disagreement. To do so, the authors propose to solve
$$ \max_{S, G} Q(S,G) \mbox{ s.t. } P(X \in S) \geq \beta, $$
which requires that the region have probability at least $\beta$ (a tuning parameter to be specified). The authors develop an iterative algorithm to optimize this objective function, and provide some conditions under which they are able to derive a generalization bound on the solution return after the first iteration (Assumptions 2-4 and Theorem 2).

## Empirical Applications
The authors apply their proposed algorithm in two empirical applications.

First, the authors analyze the behavior of their proposed algorithm in a semi-synthetic experiment that is calibrated to data from Lin et al. (2020) on recidivism risk predictions made by MTurk workers. The authors simulate data from two types of agents that follow different decision rules for detention vs. release. The authors compare their algorithm against direct models that fit logistic regressions for the decision on agent identifiers and observable features (direct models) and TARNet. The authors report that their algorithm does better at recovering the "region AUC."

Second, the authors analyze an observational dataset on medical treatment decisions for diabetes. The data contain information on 3,956 patients and 458 group practices, and the authors search for the region of disagreement across these group practices.


**Ethical Concerns:**

I have no ethical concerns with this work.

**Limitations And Societal Impact:**

Please see above for my comments on limitations and suggestions for improvement.

I do not see any potential negative societal impact from this work.

**Main Review:**

Overall, I found this paper to be interesting. The authors study a potentially important, policy relevant problem -- in many empirical settings such as the criminal justice and health care, it is useful to understand how much disagreement there is across decision makers and what types of cases have the most disagreement.

## Significance

My main concern with the paper surrounds the authors' proposed measure of disagreement across agents.

1) As mentioned above, the authors' proposed measure of disagreement ("conditional relative agent bias") measures the difference in the decisions of some fixed agent a against the average decision taken under the agent assignment mechanism.

This leads to strange properties when overlap fails. Consider a simple example where $X = \{0, 1\}$ and there are only two agents. Suppose the agents always make opposite decisions $Y(a_1) \neq Y(a_2)$ and $P(A = a_1 | X = 0) = 1, P(A = a_2 | X = 1) = 1$. In this case, the conditional relative agent bias is zero. This occurs because there is no overlap across the two agents.

Of course, as the authors point out an overlap assumption, this is not identified. But clearly this is a DGP in which we would naturally say there is a lot of disagreement across agents. However, their estimand would there is no disagreement across agents.  I think this deserves more discussion in the paper.

2) The authors' proposed measure of disagreement is also not signed. It would appear to be more natural to consider an absolute conditional relative agent bias? In this way, disagreement in any direction contributes to overall disagreement. In contrast, with this current definition, it seems like you could have a setting where one group of agents is more likely to choose Y = 1 on average, and a second group of agents is less likely to choose Y = 0 on average but these positive vs. negative biases of the groups could cancel each other out. Is this not possible? If it is possible, why should we not be worried about understanding disagreement with this objective?

3) The authors aggregate their proposed measure of disagreement into the estimand $Q(S, G)$. Again, the fact that the authors use the agent assignment mechanism leads to some strange behavior here. In particular, it implies that for any $S$, $Q(S, G) = 0$ if $G$ is equal to the set of all agents. This can be most clearly seen from their identification result in Theorem 1. If $G$ always equals 1, then
$$E[(Y - E[Y | X]) G \mid X \in S] = E[(Y - E[Y | X]) \mid X \in S] = E[Y \mid X \in S] - E[Y \mid X \in S] = 0$$.
This is also strange since it implies that disagreement is zero whenever G is chosen to be all possible agents. That's not an intuitive property of a measure of disagreement and deserves some discussion.

Since this estimand is the core object of the paper, it is essential for the authors to clearly motivate it, and discuss its limitations. The authors have not sufficiently done so in this paper. I have other comments about the significance of these results as well:

4) The authors assume that agents are assigned to cases as-if randomly conditional on the features $X$. In many empirical applications of interest, such as the cited examples in the pretrial release system (e.g., Arnold, Dobbie & Yang 2018; Kleinberg et al. 2018 which the authors cite), decision makers are \textit{quasi-randomly} assigned to cases conditional on some location-by-time cells (i.e., judges working in a given courthouse/county on a given day of the year are as-if randomly assigned to cases). Denote these location-by-time cells by $T$. The typical assumption in empirical applications is $A \perp (Y(a), X) | T$, which of course implies that $A \perp Y(a) | T, X$.

Under this assumption, the authors methods search for the region of heterogeneity over both (T, X). Can this be extended to only search for a region of heterogeneity that only depends on X? For example, policymakers may be interested in only measuring heterogeneity across defendant characteristics but not across the court-by-time cell itself.  Discussing this case would enhance the empirical relevance of these methods.

5) The authors emphasize that avoiding the overlap/positivity assumption is a key contribution of the paper.

Is this a common problem in practice? For example, if decision maker's are totally randomly assigned (for example, in my comment #3 imagine focusing on a single court-by-time cell), then A \perp (Y(a), X). If P(X) > 0, then overlap is satisfied. It would be useful for the authors to provide some concrete examples (perhaps in their second empirical application?), where the failure of overlap is a serious problem.

6) The authors require strong assumptions for their derivation of the generalization error in Section 3.4 -- e.g., only two groups of agents that follow distinct policies in Assumption 2 and a positivity type condition in Assumption 4. It would be useful for the authors to succintly summarize the roles of these assumptions in the paper. Could they be relaxed in any way? Moreover, it is strange that the authors introduce a positivity type condition in their error analysis but spend much of Section 2 emphasizing that they don't need to make this assumption for the identification result. If I need to make a positivity type condition for estimation, why wouldn't I be willing to make that same assumption when I determining what is identified from the DGP?

7) Is the conditional random assignment assumption a reasonable assumption in the second empirical application to diabetes treatment? For example, if this is a primary care setting, there are many reasons we would suspect that patients are not randomly assigned to doctors -- e.g., patients may self-select to particular doctors based on information that we do not observe, doctors may have closed practices that limit the number of patients that they may see. The authors should discuss why this assumption is reasonable in this setting.

## Originality

The authors study an important question about measuring heterogeneity in the choices of decision makers. I agree with the authors assessment that this question is often overlooked in statistics and computer science. However, there is an enormous and active literature studying heterogeneity across decision makers in economics (some of which they cite such as Arnold, Dobbie, Yang 2018 and Kleinberg et al 2018).

For example, recent work such as
* Abaluck et al. (2016) ("The Determinants of Productivity in Medical Testing: Intensity and Allocation of Care")
* Currie & Macleod (2017) ("Diagnosing Expertise: Human Capital, Decision Making and Performance Among Physicians")
* Arnold, Dobbie and Hull (2020) ("Measuring Racial Discrimination in Bail Decisions)
* Chan, Gentzkow and Yu (2020) ("Selection with Variation in Diagnostic Skill: Evidence from Radiologists")
* Norris (2020) ("Examiner Inconsistency: Evidence from Refugee Decisions")
* Ulrich & Ribers (2020) ("Machine predictions and human decisions with variation in payoffs and skill")
study this same question in the same empirical settings studied by the authors (i.e., criminal justice and health care).

These papers typically place behavioral/structural assumptions on what the authors call the "potential decision" and estimate heterogeneity across decision makers using discrete choice modelling. This additional structure is often useful for domain experts since it can be leveraged to ask and answer interesting policy counterfactuals ("how would decisions change if all decision makers have the same threshold, etc.). In this sense, the authors approach is more "nonparametric" by not modelling the potential decision, but that has a cost since it is unclear how the resulting estimand they focus on is directly useful for domain experts.  Since this question is not novel relative to this large literature in economics, it would be useful for the authors to carefully discuss how their methods and approach differ from this existing research.

## Clarity

There are two places where I found the paper to be unclear:

1) In their empirical application to recidivism prediction, the authors report the "Region AUC." I think this compares the AUC of region classifier S returned by their own model (which classifies whether X is in the region) vs. the ground truth region. Is that right? If so, it would be useful to explicitly define this in the main text.

2) In Section 3.4, the authors introduce notation $S^*, G^*$ to describe the optimal $S^*, G^*$ that maximize the objective. Are these assumed to be unique?

3) I was confused why there is some notion of a true parameter $\beta$? I would have thought that $\beta$ is a tuning parameter to be set by the user, not something that would be estimated/calibrated by the data. In other words, it is up to the user to specify what the appropriate region size is.

## Update:
I read the other reviews and the authors' responses. I thank the authors for clarifying some points that I raised on the interpretation of their proposed measure of disagreement. Provided the authors make these additional clarifications in the main text, I'd like to revise my score from 4 to 6.

**Time Spent Reviewing:**

6

---

> ### Author Response · Authors · 2021-08-10
> **Response to reviewer DZe8**
>
> We thank the reviewer for their detailed comments and appreciate the opportunity to clarify a few points.  Throughout, we number our points to line up with the reviewer comments.
>
> ### Significance of the proposed measure of disagreement.
>
> **(1)** First, the reviewer gives an example where overlap fails in a very particular way: Each value of $X$ is exclusively assigned to a *single* decision-maker. In this case, no amount of heterogeneity is identifiable from the observed data.
>
> However, there is a wide gulf between "each context is seen by exactly one decision-maker" and "each context is seen by **all** decision-makers". The latter corresponds to an overlap assumption, which we view as too restrictive.
>
> In contrast, our measure is motivated by scenarios where we expect that each context $X$ has a positive probability of being seen by **more than one** decision maker, but not by **all** decision makers. In our diabetes example, for instance, we have 458 group practices, and it would seem unreasonable to assume that each practice sees every type of patient.
>
> To continue the example given by the reviewer, suppose that $X = 0$ is always assigned to either $a_1$ or $a_2$, and $X = 1$ is always assigned to $a_3$ or $a_4$. In this case, our measure captures disagreement between $a_1$ and $a_2$ and between $a_3$ and $a_4$, even though comparisons between e.g., $a_1$ and $a_3$ are impossible to make.
>
> We will include this clarification in the main paper.
>
> **(2 & 3)**  The reviewer remarks that in the aggregate objective $Q(S, G)$, it is possible for agent-specific biases to cancel out, as the conditional relative agent bias could be negative or positive for each agent.  This perhaps reflects a lack of clarity regarding the role of $G$, which we will attempt to correct here, and re-emphasize in the main paper.
>
> $Q(S, G)$ measures the disagreement between the decision-makers in the group $G(a) = 1$ and the overall average $E[Y|X]$.  In this sense, it is unsurprising that if $G = 1$ for all agents, then $Q(S, G)$ is equal to zero:  If $G$ includes all agents, then the average of those agents is the same as the global average.
>
> This is why, in line 7 of Algorithm 1, we take $G(a) = 1$ when the conditional relative agent bias is positive, as this partially optimizes the objective $Q(S, G)$, which can then be written as
>
> $$
> \\sum_{a \\in \\mathcal{A}} {| E[Y - E[Y|X] \\mid X \\in S, A = a] |}_{+} P(A = a | X \\in S),
> $$
>
> where $|\\cdot |_{+}$ denotes the positive part.
>
> This is equivalent (up to a factor of 2) to an objective where we sum the absolute values of each agent's bias; In particular, this follows from the fact that we can write (ignoring the particular set $S$ for simplicity)
>
> $$
> \\sum_{a \\in \\mathcal{A}} ({| E[Y - E[Y|X] \\mid A = a ] |}\_{+} + { | E[Y - E[Y|X] \\mid A = a] | }\_{-} ) P(A = a ) = 0,
> $$
>
> where $|\\cdot |_{-}$ denotes the negative part.  We will clarify these points in the main paper.
>
> ### Other significance concerns
>
> **(4)** Theorem 1 would then hold for any subset $S$ of $(\mathcal{T}, \mathcal{X})$, so we are free to consider maximizing this objective only with respect to $X$ and not $T$.
>
> **(5)**  The reviewer asks for an example of when overlap is not satisfied. In our real-world experiment, in the first region of disagreement (eGFR below 58.5--see line 284), 315 of 458 group practices do not have patients in that region, yet our algorithm can still conclude that this is a region of disagreement among the 143 agents with cases in that region.
>
> **(6)** Regarding the role of these assumptions in the paper: The assumptions are limited to the analysis in Section 3.4, and the results in this section are meant primarily for intuition: When all disagreement between agents is limited to a specific region, how well does the algorithm perform, and what makes the problem harder/easier?
>
> The roles of the assumptions are as follows, as can be seen from the proof of Lemma 2 in the supplement:
> * The agent-specific bias on the region $S^*$ is lower-bounded in magnitude by $\alpha$ for each agent, by Assumptions 2 (first part) & 3, and all variation is contained within this region by Assumption 2 (second part).
> * Each agent has some positive probability of seeing contexts $X$ in the region of variation. This is the role of Assumption 4, to ensure that $P(X \in S^* \mid A = a) > \beta \omega$.
>
> All together, these assumptions imply that each agent has a positive or negative bias across all of $X$, driven by their positive or negative bias within the region $S^*$, and that this bias is large enough that we can correctly estimate the sign with some non-trivial probability during the first iteration of the algorithm.
>
> With this in mind, the assumptions can certainly be relaxed, at the expense of some additional notational complexity.
>
> For instance, the positivity-type condition (Assumption 4) can be replaced by the assumption that $P(X \in S^* \mid A = a) > \beta \omega$, which only requires that each agent sees some contexts $X$ in the region of variation (but does not require that $p(x \mid a) > 0$ for all $x \in S^*$).
>
> The proof of Lemma 2 would only change slightly: The equation prior to Eq (12) would be modified to take the integral over the set $S' = \\{ x: x \\in S^* \\land p(x | a) > 0 \\}$, then Eq (12) would involve lower-bounding $E[Y - E[Y|X] \\mid X = x, A = a]$ by $\\alpha$, and Eq (13) would proceed by observing that $\\int_{S'} P(X = x \\mid A = a) dx = P(X \\in S^* \\mid A = a)$, which can be lower-bounded by $\beta \omega$.
>
> **(7)** Regarding the plausibility of conditional random assignment (with respect to providers in the health example), we carefully considered how to ensure that unobserved confounders that affect both provider assignment and treatment decision are unlikely to occur: We included attributes that are important in existing treatment guidelines (see lines 275-278). We filtered out cases of gestational diabetes that differ from general type 2 diabetes. Finally, all patients in our dataset are from the same relatively small geographic area covered by the insurance company.
>
> ### Originality
> We thank the reviewer for the references - our work differs from the cited work in a few ways, which we will discuss here and include in the next iteration of the paper.  Throughout, we refer to the referenced work by the same citation order as given by the reviewer, [1-6].
>
> First, we note that [3] is something of an outlier, as their main focus is on a different problem: examining racial disparities in bail decisions, not disagreements between individual judges.  Among the remaining, [5] relies on having access to decisions from **multiple** decision-makers for the **same** instance;  This is feasible in their setting (multiple rounds of decisions for refugee claims) but we focus on settings where this is not feasible (e.g., there is only one decision, and hence only one decision-maker, for each patient).
>
> For the remaining references [1,2,4,6], our work differs in two main ways:  First, all of these methods require estimating a separate model for each individual agent. This is difficult to do reliably in the setting we consider, where e.g., the average number of samples per agent is fewer than 9  (see line 38).   Second, we are primarily motivated by settings in which the “correct” decision is not obvious as in our real-world experiments - in contrast to [1, 4, 6] where the decision-problem is diagnostic in nature and one can estimate measures of physician performance (e.g., false-negative rates for detecting pneumonia in [4]).
>
> Of these, closest to our work is [2], in which an aggregate logistic choice model (for C-sections) is estimated across all physicians, and the authors then learn how individual physicians deviate from this model.  However, their model does not allow for learning the regions in which this deviation occurs:  Rather, they learn each physician’s choice model as a linear function of the logits of the original model, with a slope and intercept term.  This is useful for their setting (understanding how heterogeneity is associated with downstream outcomes), but less so in ours.
>
> We will include discussion on all of these points in the paper, and we thank the reviewer for bringing these references to our attention.
>
> ### Other clarifications
>
> * Region AUC: This interpretation of the “Region AUC” is correct, we will make this more explicit in the paper.
> * $S^*, G^*$:  We will revise Section 3.4 to clarify this point - In retrospect, it would perhaps be clearer to write the assumptions in terms of a set $S'$ and a grouping $G'$, and then show (as we do in Appendix A.4) that this pair maximizes the objective, justifying the notation $S^*, G^*$.  Under these particular assumptions, they are also a unique solution.
> * True parameter $\\beta$:  We include discussion on this point to assist users in selecting a value of $\\beta$ if they do not already have a strong reason to prefer some value over another.

---

### Official Review · Reviewer_v34w · 2021-07-15

**Rating:** 7
**Confidence:** 4

**Summary:**

Summary:

This paper considers settings where decisions for individuals are each made by different decision makers (agents). The paper defines the problem of finding a region in feature space in which the agents administer highly varying decisions. For example, perhaps judges mete out different sentences for misdemeanors by young men. The paper defines new causal criteria to characterize such regions in the feature space, where the choice of agent has an effect on the decision. The paper introduces an iterative algorithm to maximize this criteria to identify regions of heterogeneity, and provides theoretical results about the optimality of the proposed algorithm. The paper empirically demonstrates the algorithm on a semi-synthetic dataset and real clinical data about doctor recommendations for diabetic patients.

Contributions:

+ A new criteria for defining regions where there is heterogeneity in decisions administered across agents.
+ Results for estimating the criteria from observational data and an iterative algorithm for optimizing the criteria.
+ Theoretical results about the algorithm and empirical validation of the algorithm.

**Limitations And Societal Impact:**

There isn't much a discussion about the limitations of this method, which I would've liked to see. This is a room for improvement.

**Main Review:**

Originality:
This paper presents an original estimation problem and algorithm to solve it. To the best of my knowledge, this paper provides a new formalization of the problem of wanting to learn about decision making heterogeneity and proposes a new criteria for measuring this heterogeneity based on regions of input space in which agents differ in their decision making.

The iterative algorithm is a reasonable optimization procedure and the theoretical results are obviously new for this algorithm, but probably borrow ideas from the optimization literature (I haven't checked the proof.)

Quality:

Although I haven't checked the proofs for the theorems, the results pass the sniff test. I looked at the algorithm closely to follow along, and there are again no surprises there, which is good.

At a high level, my one complaint is about the beginning about section 2 and Eqn 1. I feel that causality is almost incidental to the paper because of the way the estimand in Eqn 1 is defined. Indeed, as the paper says, the challenge of causal inference is if we needed to evaluate Y(a') for a person who was assigned A=a and got Y(a) = Y|A=a. But in Eqn 1, we don't consider the impossible counterfactual Y(a') because we contrast the observed outcome with a different quantity, the average decision over agents.

Eqn 1 could be rewritten as:
$$\mathbb{E}[Y|A=a, X \subset S] - \mathbb{E}[Y|X \subset S]$$
The interpretation of the agent bias criteria still holds: we can think of it as the difference between a's decision and the expected decision across agents for that region of feature space. This formulation also connects the algorithm and criteria to the "direct" baseline.

Mainly, I wonder if invoking causality is an unnecessary distraction in this paper. Can the authors explain why we need to consider counterfactual notion from their view?

Another technical point where I have a doubt is: how does this algorithm fare when we have a high dimensional feature space? In the empirical studies, both settings consider just a few features. Is that because the algorithm fails with high dimensional data, or is it for ease? Theorem 2 does show a positive dependence between the "size" of S and the error term -- does this turn out to be a practical hindrance, restricting the algorithm to small feature sets?

I think the empirical studies are thorough, with both a semi-synthetic study and a real world case study where we have validation based on domain knowledge. I think especially given that the authors had to come up with their own baselines, they did a good job of designing comparisons.

However, I would've liked a better explanation about the motivation behind using TARNet baseline as its used. That is, what's the significance of predicting E[Y|X,A] and then calculating Var[E[Y|X,A]]?

Finally, do the authors have a hypothesis for why Direct performs comparably with Iterative in the misdemeanor setting?

Clarity:
The paper is reasonably well written, with enough detail around the algorithm and experimental evaluation for a reader to implement the algorithm themselves or to understand the studies.

The authors also provided good plain English explanations about the agent bias criteria that they defined, which helped build intuition. There was also exposition to give intuition about the algorithm.

However, I think that the notation of the form $\mathbb{E}[Y | X \subset S]$ is unconventional and needs a clear, mathematical explanation prior to equation 1. Initially, I interpreted it as E[Y|X], which is a random variable in X, and I got extremely confused with the equivalence in Eqn 2, which clearly evaluates to a scalar.

I might even give Eqn 2 right away without Eqn 1 with more explanation about the notation. The intuitions that came after Eqn 2 were important for understanding the interpretation of the quantity.

Significance:

The paper formalizes the problem of decision making heterogeneity and offers a practical procedure to find regions where agents make different decisions. I think this offers a significant contribution to domain experts as well as researchers in causal inference who will want to build upon this work.


**Time Spent Reviewing:**

4 hours

---

> ### Author Response · Authors · 2021-08-10
> **Response to Reviewer v34w**
>
> We thank the reviewer for the detailed review and appreciate the opportunity to clarify a few points.
>
> **The need to consider counterfactual notation**:  It is not enough to observe that different agents make different decisions on average - rather, we would like to attribute these differences to meaningful variation in preferences (or other agent-specific factors).
>
> Our causal formalism allows us to clearly state the assumptions required to do so. For instance, suppose that all doctors follow the same policy, using more aggressive treatment for more severely ill patients, but that **severity of illness is not captured in $X$**.  As a result, doctors with more severely ill patients will appear to “prefer” more aggressive treatment, even if all doctors follow the same exact policy.  This would correspond to a violation of the “no-unmeasured-confounding” assumption.
>
> As a minor correction to the reviewer’s comment: Equation (1) cannot be re-written as suggested.  Rather, the correct re-writing of Equation (1), under the causal assumptions in Assumption 1, is the following
> $$E[Y \\mid A = a, X \\in S] - E[ E[Y|x] | A = a, X \\in S],$$
> where the second expectation is taken over $x$, recalling that $E[Y|x]$ is a function of $x$.  In particular, the second expectation is taken over the distribution $P(X = x \\mid A = a, X \\in S)$. This is subtly different from the suggestion of the reviewer
> $$E[Y \\mid A = a, X \\in S] - E[Y \\mid X \\in S],$$
> in that the former captures the fact that $A$ may tend to see a particular subset of $X$ values within $S$.  For a simple example, suppose that $S$ is the set of elderly patients;  It may be the case that doctor $A=a$ sees elderly patients who are healthier than the average elderly patient, a nuance that is not captured by only considering $E[Y \\mid X \\in S]$
>
> **Scaling to high dimensions**:  While we do not have quantitative evidence to offer (since we did not explore this in our experiments), there is no particular theoretical reason to believe that the method would suffer in high dimensions, aside from the usual difficulties of fitting conditional predictive models f(x) and h(x).  The other parts of the algorithm (e.g., estimating groups in line 6 of Algorithm 1 via simple averages) have no explicit dependence on dimension.  Regarding the “size” of the set S, this is best described by the Rademacher complexity, which depends on the complexity of $h(x)$.
>
> **Motivation behind TARNet baseline**: Regarding the baselines, the reviewer raises a question regarding the motivation behind our choice of the TARNet baseline. We use TARNet as a baseline because it directly estimates the expected decision for each agent, providing counterfactual decisions we can compare to identify a region of disagreement. This baseline does not provide an intuitive way to group the agents. Thus, $Var\_A[E[Y | X, A]]$ measures the variation across all agents if they had seen context $X$. This becomes the objective we maximize when identifying the region. We will add these clarifications to our paper.
>
> **Direct baseline in misdemeanor setting**: In the right subplot of Figure 1, it appears that the direct baseline performs almost as well as our iterative algorithm in the misdemeanor set-up.  However, we note that there is a small mix-up here:  Per the caption, this subplot is supposed to show the comparison to the iterative algorithm with random forests (the best-performing region model), but instead shows the comparison to the iterative algorithm with ridge regression - the correct comparison is given in the top right subplot of Figure 4 in the supplement.  We hypothesize the direct baseline does almost as well as our iterative algorithm with the ridge regression region model because the region cannot be represented by a ridge regression.
>
> **Clarity**: We will also make sure that the notation $\\mathbb{E}[Y \\mid X \\in S]$ is clearly explained, via a simple example:  In particular, if we wrote $\\mathbb{E}[Y \\mid X < 10]$, one would see that this is a scalar - the average of $Y$ among samples where $X < 10$, where in our notation the set $S$ would consist of all $X$ such that $X < 10$.
>
> **Limitations**: We do identify limitations of our algorithm in lines 31-33 of the introduction and the causal identification assumptions in lines 68-69, but we will expand upon these limitations to reflect the discussion here (for instance, the lack of quantitative evidence that our method scales to higher dimensional problems).

---

> > ### Comment · Reviewer_v34w · 2021-08-22
> > **Thanks for the detailed response**
> >
> > Authors, thanks for taking the time to answer my questions and provide clarifications. I see your point about why the counterfactual notation is needed and isn't just a distraction here.

---

### Official Review · Reviewer_9a1j · 2021-07-22

**Rating:** 9
**Confidence:** 3

**Summary:**

This paper proposes a new method to identify "regions of disagreement" in a setting with human decision-makers assigned to individuals (eg. judges assigned to court cases). The "regions" are framed as subgroups of the data (eg. males over 30 years old) which can be interpretable (based on the chosen model). The paper formalizes this task as that of identifying the subgroup where the identity of the specific decision-maker has a causal effect on the decision, proposes an optimization algorithm to find this subgroup, develops generalization bounds (under some simplifying assumptions), and demonstrates compelling performance on semi-synthetic and real-world datasets.

**Limitations And Societal Impact:**

**Potential negative societal impact.** As a method of "discovery", there is is always the potential for false discoveries with bad consequences (especially when no statistical inference safeguards are provided). In addition, the models f(x) and h(x) could propagate biases in the data they are trained on, which might spill over in to the regions discovered in unexpected ways. The authors don't actually talk about the potential negative societal impacts in Section 1 (they only state that Assumption 1 is a limitation). It is essential that these concerns be mentioned early in the paper.

**Main Review:**

## General Points

Finding regions of "disagreement" or "heterogeneity" in the decisions made by human experts is important to diagnose decision-making issues. For example, if we find that judges are highly variable in the decisions they make for young males, it might be worth creating a uniform judgement policy for such males to improve judgement consistency, and reduce the potential impact of personal biases. If we find that teachers grading essays tend to disagree on the scores assigned to female students, it might be worth hiding student identities. In both cases, before enacting any policy changes, **we need a principled method to identify such regions of heterogeneity, which this paper provides.**

This specific task falls under the broader umbrella of *subgroup discovery*. One of my favorite recent examples of work in this space is [this paper](https://arxiv.org/pdf/1707.00046.pdf) on identifying regions/subgroups where 2 machine learning models (not humans) disagree (with a specific focus on disagreement  in fairness). This paper is also related to work on *heterogeneous treatment effect estimation* (the paper on [causal forests](https://www.tandfonline.com/doi/abs/10.1080/01621459.2017.1319839) is a canonical example of this space). **Unlike the aforementioned 2 papers, this paper does not use recursive partitioning and has a clever, novel formulation of the subgroups** (those where the causal effect of the assigned decision-maker is non-zero). I believe this formulation itself will be useful for the fairness literature, and that this paper can benefit from some of the ideas in the aforementioned 2 papers.

I have summarized the key strengths and weaknesses of this paper below.

## Key Strengths

**Important problem and clever, novel formulation.** I have stressed the problem importance and novelty above. I want to note that the proposed formulation and optimization method is fairly flexible and can be made interpretable and tailored to specific domains. This helps expand the proposed method's applicability to a wide range of real world settings.

I must also stress that the proposed causal estimand in eq. 2 is non-trivial to come up with: it is typically hard to come up with a causal estimand for a new task that makes intuitive sense, that is identifiable from observable data, and that is estimable from data in an efficient manner.

In addition, Theorem 1 transforms eq. 3 which contains the propensity scores P(a|s) into eq. 4 that does not require knowing these propensity scores. What this means is that, as long as the (conditional) randomization/no unobserved confounders is satisfied, we do not need to know the specific randomization protocol! This enables applying this method to settings like judges assigned to cases, and physicians assigned to patients, where we know that there is randomization but the specific assignment mechanism is proprietary/unknown. Without Theorem 1, this method would probably be restricted to randomized control experiments where the decision-maker assignment probabilities are known.

**Clear and complete description of the causal task.** This paper states the causal estimand, the identification assumptions, and derives an identification proof to link unobservable counterfactual quantities to observable ones. As such, the description of the causal task is clear and complete. Importantly, the mathematical quantities are also described intuitively and linked to real-world quantities, which aids readers interested in applying this method to their setting.

**Compelling synthetic and real world performance.** The paper empirically demonstrates performance in a well-described collection of experimental settings, one synthetic and two real world. The method is evaluated along several dimensions, and recipes are provided for essential real-world tasks such as interpreting and assessing the significance/stability of the identified region

## Key Weaknesses

**The proposed method finds the one region with maximum heterogeneity.** In practice, we can expect there to be multiple regions where the agents disagree; the proposed method would not help identifying all such regions. While it might be tempting to simply tweak the proposed method to find all such regions, it would be very difficult to address issues of multiple testing / false discovery (if we test enough regions of a large feature space, we will likely find at least a few that appear to have significant heterogeneity). The paper might want to clarify this in the problem definition, since the title is a bit misleading in this regard.

**It is unclear how the method would perform and scale as the dimensionality of the feature space increases.** I expect that, with high dimensional $X$ and several irrelevant features, it would be harder to find valid regions of heterogeneity. In addition, I wonder how the running time / no. of iterations of the optimization algorithm scale with the size of the feature space, no. of decision makers, and no. of data samples.

**While the no unobserved confounders and consistency assumptions are stated, its real world implication need to be clarified up front.** What would violate these assumptions in the real world (provide concrete examples)? It is very important to clarify, for future readers of this method, that you **need (conditional) randomized assignment of decision-makers for this method to work.** This is not a dealbreaker, but I can imagine future readers brushing over the no unobserved confounders assumption without realizing how restrictive it is.

**While recipes to assess significance/stability are provided, it is unclear if these are statistically valid.** Ideally, one would have a way to construct "confidence intervals" around the identified region of heterogeneity. While this paper provides a recipe to test significance, I am not sure how statistically valid that is (can validity be demonstrated empirically?). The paper needs to caution readers about the potential for false discoveries without a statistically valid inference procedure.

**Concerns with estimating the f(x) and h(x) models on the same data where the causal effect is estimated.** In the "double machine learning literature" where ML models are using along with causal inference, the ML models are estimated on a *separate sample* of the data. This is to disentangle the errors of the causal effect estimation and the errors arising from (i) overfitting the ML models, and (ii) underfitting the ML models. This paper does not consider what happens if f(x) and h(x) are poor models (does it lead to an increased or lowered risk of false discovery of regions of heterogeneity?).

**Intuition behind the grouping objective in eq. 3.** The grouping mechanism was introduced without much intuition; could the equation be supplemented with a few descriptive lines? I believe because of the grouping, there is no need to place an absolute value around eq. 3, but I could be wrong.

**Optimization guarantees.** Does the optimization method find a global optimum (I know most subgroup discovery methods are greedy and do not find the globally optimal region of heterogeneity, but it might be good to state this clearly to readers)?

**Potential negative societal impact.** As a method of "discovery", there is is always the potential for false discoveries with bad consequences (especially when no statistical inference safeguards are provided). In addition, the models f(x) and h(x) could propagate biases in the data they are trained on, which might spill over in to the regions discovered in unexpected ways. The authors don't actually talk about the potential negative societal impacts in Section 1. It is essential that these concerns be mentioned early in the paper.

**Time Spent Reviewing:**

12

---

> ### Author Response · Authors · 2021-08-10
> **Response to Reviewer 9a1j**
>
> Thank you for the detailed review and highlighting the contributions of our paper.
>
> **Multiple regions**: Our method can find several disjoint regions as part of the region of heterogeneity $S$. See Figures 9-10 in the supplement for examples in our real-world experiments.  When we mentioned multiple “regions” in the title, this is what we had in mind, but we will clarify this point in the paper.
>
> **Scaling with dimensionality of $X$, # of decision-makers, etc**:  There is no particular theoretical reason to believe that the method would suffer in high dimensions, aside from the usual difficulties of fitting conditional predictive models $f(x)$ and $h(x)$ in higher dimensions. However, we do not have quantitative evidence to offer, as we did not explore this in our experiments.  This is certainly a reasonable practical concern, and we will make note of it as a possible limitation.
>
> We did explore scaling with # of decision-makers - Figures 5 and 6 in the supplement show that our algorithm scales well as the number of agents increases (10, 20, 40, and 87). Convergence happens in a few iterations. Similarly, our real-world experiment had 458 agents (see line 280), and our algorithm ran in a few seconds.
>
> **Real-world implications of assumptions**:  We will add concrete examples to the paper of when these assumptions may not hold:  For instance, if gestational diabetes cases are included without pregnancy as a feature, then this would create unobserved confounding: Gestational diabetes patients may see different providers and receive different treatments, but this variation is not due to differences in provider policies, merely their mix of patients.
>
> **Statistical tests**:  While we provide sanity checks as noted by the reviewer, we do not provide formal statistical tests (e.g., that no variation exists), a limitation that we will make clearer in the next iteration.
>
> **Concerns with $f(x)$ and $h(x)$**: When fitting $f(x)$, we use held-out validation data to determine if the model is well-fit.  We appreciate the suggestion regarding sample splitting, though this leads to some risk of under-fitting in $f(x)$, due to the already small sample sizes in some of our motivating applications.  This could lead to false discovery, if $f(x)$ does not capture all the variation that can be explained by $X$.  We will add discussion on this point to the paper.
>
> **Intuition behind grouping objective**:  We go into more depth regarding the intuition behind the grouping objective in Equation (3) in our response to Reviewer DZe8 (see our response to their points 2&3);  In particular, because we maximize this objective wrt $G$, we do not need to include absolute values in the equation.  We will add some of this discussion into the main paper, as requested.
>
> **Optimization guarantees**: We make no claim that the iterative algorithm will find a globally optimal solution, and we will clarify this potential limitation to readers.
>
> **Societal impact**:  We thank the reviewer for raising these points - we will expand upon our existing discussion of caveats in the introduction (lines 31-33) to make note of these points as possible negative societal impacts.
>
> **Causal forest**: As a side note, the causal forest paper the reviewer cited is adapted into one of our baseline methods (see lines 706-721 in the supplement).

---

> > ### Comment · Reviewer_9a1j · 2021-08-31
> > **Re: Response to Reviewer 9a1j**
> >
> > Thank you for your response, it addresses my concerns and I hope the changes promised in the response do make their way into the final paper.

---

### Decision · Program_Chairs · 2021-09-27

**Decision:**

Accept (Poster)

**Comment:**

In this work the authors propose a method for identifying regions (types of instances) where there is a high level of heterogeneity in in human decision making.  Their approach is grounded in a causal inference framework that models high-heterogeneity regions as ones where the choice of decision maker ("treatment") has a large effect on the resulting decision.

This paper received a number of excellent high-quality reviews that raised important concerns and requests for clarification.  The authors responded well to the major concerns raised by the different reviewers and have clearly articulated how they will revise the manuscript to improve clarity and further contextualize the work within a broader set of relevant literature.  I agree with the reviewers that this paper tackles an important problem and that the work meets the bar both in terms of technical contribution and exposition.  In revising the manuscript, the authors should seek to respond to as many of the reviewer suggestions as possible, with an emphasis on:

- Adding and briefly discussing the related work suggested by reviewers
- Making assumptions more tangible by giving clear examples of when they may or may not hold in practice
- Clarifying the particular type of overlap assumption upon which the work relies, and contrasting with all-agent overlap
- Clarifying why agent-specific biases do not cancel out in the aggregate objective